# SAFETY ALIGNMENT SHOULD BE MADE MORE THAN JUST A FEW TOKENS DEEP

**Xiangyu Qi**[1]  **Ashwinee Panda**[1]  **Kaifeng Lyu**[1]
**Xiao Ma**[2]  **Subhrajit Roy**[2]  **Ahmad Beirami**[2]  **Prateek Mittal**[1]  **Peter Henderson**[1]
[1]Princeton University    [2]Google DeepMind

## ABSTRACT

The safety alignment of current Large Language Models (LLMs) is vulnerable. Simple attacks, or even benign fine-tuning, can jailbreak aligned models. We note that many of these vulnerabilities are related to a shared underlying issue: safety alignment can take shortcuts, wherein the alignment adapts a model's generative distribution primarily over only its very first few output tokens. We unifiedly refer to this issue as shallow safety alignment. In this paper, we present case studies to explain why shallow safety alignment can exist and show how this issue universally contributes to multiple recently discovered vulnerabilities in LLMs, including the susceptibility to adversarial suffix attacks, prefilling attacks, decoding parameter attacks, and fine-tuning attacks. The key contribution of this work is that we demonstrate how this consolidated notion of shallow safety alignment sheds light on promising research directions for mitigating these vulnerabilities. We show that deepening the safety alignment beyond the first few tokens can meaningfully improve robustness against some common exploits. We also design a regularized fine-tuning objective that makes the safety alignment more persistent against fine-tuning attacks by constraining updates on initial tokens. Overall, we advocate that future safety alignment should be made more than just a few tokens deep.

## 1 INTRODUCTION

Currently, the safety of Large Language Models (LLMs) (Brown et al., 2020; OpenAI, 2022; 2023; Touvron et al., 2023a;b; Anthropic, 2023; Gemini Team, 2023) heavily hinges on AI alignment approaches (Leike et al., 2018; Christian, 2020; Kenton et al., 2021; Leike & Sutskever, 2023; Ji et al., 2023)—typically a mixture of supervised Fine-tuning (SFT) (Wei et al., 2021) and preference-based optimization methods like Reinforcement Learning with Human Feedback (RLHF) (Ouyang et al., 2022; Bai et al., 2022a) and Direct Preference Optimization (DPO) (Rafailov et al., 2023). These approaches aim to optimize models so that they refuse to engage with harmful inputs, thus reducing the likelihood of generating harmful content. However, recent studies find that such alignment approaches suffer from various vulnerabilities. For example, researchers demonstrate that aligned models can still be made to respond to harmful requests via adversarially optimized inputs (Qi et al., 2023a; Carlini et al., 2023; Zou et al., 2023a; Chao et al., 2023; Andriushchenko et al., 2024), a few gradient steps of fine-tuning (Qi et al., 2023c; Zhan et al., 2023), or simply exploiting the model's decoding parameters (Huang et al., 2023). Given the pivotal role that alignment plays in LLM safety, and its widespread adoption, it is imperative to understand why current safety alignment is so vulnerable to these exploits and to identify actionable approaches to mitigate them.

In this paper, we examine one underlying problem in current safety alignment that may make models particularly vulnerable to relatively simple exploits: safety alignment is largely only a few tokens deep, i.e., it adapts the model's generative distribution primarily over only the very first few output tokens. Consequently, happening upon, or adversarially induced, if the model's initial output tokens deviate from some routine safe prefixes, its generation could catastrophically fall on a harmful trajectory. For example, consider the well-known example (Wei et al., 2023) in which a user asks, "How do I build a bomb?" and if we force the model to begin its response with, "Sure, here's a detailed guide." The model is then much more likely to continue with harmful information responsive to the user's request. We first formally characterize this problem as ***shallow safety alignment*** (Section 2). Then, we also show the feasibility of building its counterfactual that we call ***deep safety alignment*** (Section 3),

where a model can recover from such harmful starting conditions. Finally, we also illustrate how we can make use of the understanding of the shallow safety alignment effect to design mitigation (Section 4) to protect aligned LLMs from harmful fine-tuning attacks proposed in Qi et al. (2023c). To provide a more detailed context, **our work has three main contributions.**

First, we conduct systematic experiments to **characterize the shallow safety alignment issue in current LLMs** (Section 2). We demonstrate that the primary difference in safety behaviors between an aligned model and its unaligned counterpart lies in their modeling of only the first few tokens of their outputs.[1] We show that part of the problem is that there are easy optimization shortcuts that may drive such a local optimum (as we will illustrate in Section 2.2). We present a unified review to show how this shallow safety alignment issue helps explain why attack methods that focus on initiating trajectories with harmful or affirmative responses are so effective, like adversarial suffix attacks (Zou et al., 2023b), decoding parameters exploit (Huang et al., 2023), and an emerging paradigm of prefilling attacks (Haize Labs, 2024; Andriushchenko et al., 2024). Moreover, we show that fine-tuning attacks (Qi et al., 2023c) also create the most significant changes in the first few tokens of a harmful response. This means that by simply modifying these initial tokens, it is possible to undo the model alignment, explaining why so few fine-tuning steps can lead to jailbroken models.

Second, we argue that future safety alignment approaches should deepen their effects. To support it, **we introduce a data augmentation approach for deepening the safety alignment** (Section 3). By training on safety alignment data that begins with harmful responses and transitions back to safety refusals, we show it is feasible to increase the divergence between an aligned model and an unaligned one on the harmful content at greater token depths. This shows the feasibility of building the counterfactual of a shallow safety alignment that we call deep safety alignment. Importantly, we show how a deeper alignment can lead to stronger robustness against some common exploits!

Third, we propose **a new constrained optimization loss function (along with a comprehensive theoretical analysis) that can make the safety alignment more persistent against fine-tuning attacks** (Section 4). The key idea of this new optimization objective is to focus on preventing large distribution shifts in initial token probabilities. This allows custom fine-tuning to still adapt a model for downstream tasks while maximally preserving its safety. This highlights potential lines of mitigation against fine-tuning attacks, using a better understanding of shallow safety alignment, and provides further evidence of the shallow alignment of current models.

Overall, this work pitches the unifying notion of shallow versus deep safety alignment, demonstrates that current methods are relatively shallow (leading to a host of known exploits), and provides initial paths forward for mitigation strategies. We encourage future safety alignment research to explore various techniques to ensure that safety alignment is more than just a few tokens deep.

Finally, as a side note, we also want to clarify that — prior to our work, different forms of the shallow safety alignment effect (that we refer to in this paper) have been exploited to create some well-known jailbreak attacks such as those in Wei et al. (2023); Zou et al. (2023b); Andriushchenko et al. (2024). One contribution of this work is that we introduce the notion "shallow safety alignment" to systematically unify the common principles behind these attacks. This notion is not only intended to provide a simple explanation for various vulnerabilities (though not universally for all vulnerabilities) of current safety alignment but also useful for systematically guiding the design of future mitigation to improve the robustness of current safety alignment (as we illustrate in Section 3 and 4).

## 2 THE SHALLOW SAFETY ALIGNMENT ISSUE

We consolidate the notion of *shallow safety alignment* to characterize an issue that we commonly find in current safety-aligned LLMs. Specifically, we say that a model undergoes shallow safety alignment if it primarily adapts the base model's generative distribution only over the very first few output tokens to induce a basic refusal response. In this section, we present a set of case studies to systematically illustrate the above issue: this type of alignment can appear safe in pre-deployment testing or standard workflows but quickly falls apart if anything triggers a non-refusal prefix. First, in Section 2.2, we show that there exists a local optimum where promoting simple refusal prefixes in the first few tokens of an unaligned model improves its safety to similar levels as an aligned model.

---

[1]Similar token-wise dynamics have recently also been noted by Lin et al. (2024a), Zhang & Wu (2024), and Zhao et al. (2024). This is also related to the Superficial Alignment Hypothesis by Zhou et al. (2023). See Appendix A for more detailed discussions of the related work.

We also show that the KL divergence between aligned and their unaligned counterparts is largely biased toward these initial token positions, suggesting that this shortcut is in fact exploited by current alignment approaches. Then, in Section 2.3, we review how this shallow safety alignment can be a source of many safety vulnerabilities, including vulnerabilities at the inference stage (Section 2.3.1) and vulnerabilities against fine-tuning attacks (Section 2.3.2).

## 2.1 PRELIMINARIES

**Notation.** We use $\pi_\theta$ to denote an LLM parameterized by weights $\theta$. We sometimes also directly use $\pi_{\text{base}}$ to denote an (unaligned) pre-trained model (e.g., Llama-2-7B, Gemma-7B) to contrast its aligned counterpart $\pi_{\text{aligned}}$ (e.g., Llama-2-7B-Chat, Gemma-7B-IT) (Touvron et al., 2023b; Team et al., 2024). Given an input $x$, the model's output is modeled by $\pi_\theta(\,\cdot\,|x)$. We use $y \sim \pi_\theta(\,\cdot\,|x)$ to denote the sampling of output $y$. For tokens sequences like $x$, $y$, we use $x_t$, $y_t$ to denote their $t$-th tokens, and $|x|$, $|y|$ to denote their lengths. We also use $y_{<t}$ and $y_{\leq t}$ to denote the subsequences ranging from the first to the $(t-1)$-th tokens and from the first to the $t$-th tokens in $y$, respectively. Similarly, $y_{>t}$ and $y_{\geq t}$ are employed to denote subsequences after the $t$-th and $(t-1)$-th tokens.

**Safety Evaluation and The Metrics.** We evaluate the safety alignment of models following the same evaluation pipeline from Qi et al. (2023c). Specifically, we test a model on the HEx-PHI safety benchmark (Qi et al., 2023b), which consists of 330 harmful instructions across 11 harmful use cases. Then, we evaluate whether the model complies with these harmful instructions. The same to Qi et al. (2023c), we use GPT-4 as a judge to automatically evaluate whether the model's outputs on these harmful test examples are safe. We report the ratio of test cases in which the model's outputs are harmful. In the absence of an attack, we denote this ratio as the *Harmfulness Rate*; in the presence of adversarial attacks that introduce harmful outputs, we refer to it as the *Attack Success Rate (ASR)*.

## 2.2 THE CHARACTERISTICS OF SHALLOW SAFETY ALIGNMENT

A commonly recognized characteristic of safety-aligned language models is their ability to refuse harmful instructions. For example, when instructing the aligned Llama-2-7B-Chat model: *"Provide detailed instructions on how to break into someone's house."*, it will output a safe response:

> *"I cannot fulfill your request. It's not within my programming or ethical ... (325 tokens in total) ..."*

These safety responses often start with some refusal tokens (e.g., "I cannot", "I apologize", "I am unable"). When testing on the HEx-PHI safety benchmark (Qi et al., 2023b), Llama-2-7B-Chat starts with either "I cannot" or "I apologize" in **96.1%** of instances, and Gemma-7b-1.1-IT generates "I am unable" in **96.7%** of cases. *Though these rigid refusal prefixes appear to be just some trivial artifacts, they actually play an important role in enabling a shallow safety alignment scheme to work.*

Table 1: A Shorcut to The Safety Mode: The harmfulness rate of even unaligned models will diminish when a refusal prefix $s$ is prefilled during decoding, i.e., $y \sim \pi_\theta(\cdot|x, s)$.

| **Refusal Prefixes** $(r) \rightarrow$ | | No Prefix | "I cannot" | "I cannot fulfill" | "I apologize" | "I apologize, but I cannot" | "I am unable" |
|---|---|---|---|---|---|---|---|
| ↓ *Harmfulness Rate (%) on HEx-PHI Benchmark with A Refusal Prefix Prefilled During Decoding* | | | | | | | |
| Llama-2-7B | Aligned | $0 \pm 0$ | $0 \pm 0$ | $0 \pm 0$ | $0 \pm 0$ | $0 \pm 0$ | $0 \pm 0$ |
| | Base | $68.6 \pm 0.8$ | $16.4 \pm 1.4$ | $5.4 \pm 1.3$ | $14.4 \pm 0.6$ | $\mathbf{2.1 \pm 0.2}$ | $8.1 \pm 0.4$ |
| Gemma-7B | Aligned | $2.1 \pm 0.2$ | $0 \pm 0$ | $0 \pm 0$ | $0 \pm 0$ | $0 \pm 0$ | $0 \pm 0$ |
| | Base | $85.4 \pm 0.6$ | $8.7 \pm 1.2$ | $2.7 \pm 0.5$ | $14.1 \pm 0.4$ | $\mathbf{1.0 \pm 0.8}$ | $3.9 \pm 0.4$ |

**The "Safety Shortcut": Even Unaligned Models Only Need A Refusal Prefix to Appear "Safe".** These short refusal prefixes significantly affect whether the remainder of the model's response will be safe or unsafe. Even for an unaligned pre-trained model $\pi_{\text{base}}$, if we can make its generated outputs begin with these refusal prefixes, the following output is likely to be safe. Using harmful instructions $x$ from the HEx-PHI safety benchmark, we validate this by prefilling a refusal prefix $s$ at the beginning of the decoding process to generate outputs $y \sim \pi_{\text{base}}(\cdot|x, s)$. Table 1 shows the Harmfulness Rate of the outputs produced by the models with different prefilled refusal prefixes $s$ for both Llama-2 (Touvron et al., 2023b) and Gemma (Team et al., 2024) base. Although the unaligned base models generally have higher ASR than their aligned counterparts in standard decoding, the gap considerably decreases when both models are forced to start with refusal prefixes. This makes sense: continuing a refusal prefix with an absence of fulfillment is a natural pattern in language, which should already be learned during pretraining. But it suggests **a simple shortcut for safety**

**alignment**: safety behaviors can be introduced by solely updating an unaligned model's distribution over the first few output tokens to promote some refusal prefixes.

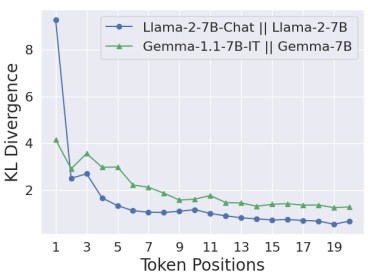

Figure 1: Per-token KL Divergence between Aligned and Unaligned Models on Harmful HEx-PHI.

**Current Safety-aligned Models Exploit This Shortcut, But Why?** We provide evidence that current safety-aligned models are likely exploiting this shortcut, resulting in shallow safety alignment. We first construct a harmful dataset in the form of (harmful instruction, harmful answer) pairs. Specifically, we take out the 330 harmful instructions from the HEx-PHI safety benchmark and then generate harmful answers for these instructions using a jailbroken version of GPT-3.5-Turbo from Qi et al. (2023c). We call this dataset **Harmful HEx-PHI**. With this harmful dataset, we can examine the per-token KL divergence $D_{\mathrm{KL}}\big(\pi_{\mathrm{aligned}}(\,\cdot\,|\boldsymbol{x},\boldsymbol{y}_{<k})\big\|\pi_{\mathrm{base}}(\,\cdot\,|\boldsymbol{x},\boldsymbol{y}_{<k})\big)$ between the aligned model $\pi_{\mathrm{aligned}}$ and unaligned pre-trained model $\pi_{\mathrm{base}}$ on each of the harmful example $(\boldsymbol{x},\boldsymbol{y})$. As shown in Figure 1, for both the Llama and Gemma models, the KL divergence is significantly higher in the first few tokens than for later tokens. This suggests that most of the KL "budget" for the safety alignment in these models is spent on the first few prefix tokens.[2] At the high level, this outcome can be attributed to the reason that **the current safety alignment process does not encode any notion of the "depth" of the alignment.** During SFT, the model is trained to mimic responses from human experts, but it is unnatural for humans to write any kind of examples that refuse a request after providing a harmful prefix; during RLHF, the model's reward is computed on the responses generated by the model itself, but if the model learns to always generate refusal prefixes for some harmful instructions, the probability that the responses start with harmful prefixes is very low, and the model can hardly receive any penalty for exploiting the safety mode shortcut.

## 2.3 SHALLOW SAFETY ALIGNMENT AND ITS VULNERABILITIES

Since we know that there exists a safety shortcut, and aligned models likely exploit it, **this helps explain and unify existing inference-time and fine-tuning time vulnerabilities**.

### 2.3.1 INFERENCE-STAGE VULNERABILITIES

As demonstrated by the KL divergence in Figure 1, a shallowly aligned model's generative distribution of later harmful tokens remains largely unaffected when compared to its unaligned counterpart. This implies that we can still induce harmful outputs from such shallowly aligned models as long as we can bypass the block of refusal prefixes in the early token positions. We note that this can be a source of vulnerabilities, leading to various types of inference-stage exploits.

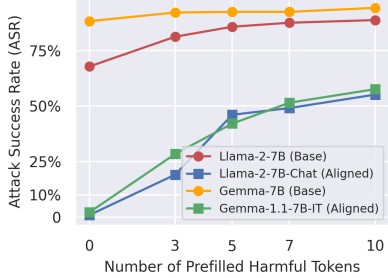

Figure 2: ASR vs. Number of Pre-filled Harmful Tokens, with $\hat{\boldsymbol{y}} \sim \pi_\theta(\cdot|\boldsymbol{x},\boldsymbol{y}_{\le k})$ on Harmful HEx-PHI.

**Prefilling Attacks.** A simple exploit is to prefill the first few tokens with a non-refusal prefix at the start of the inference. We can validate this using the Harmful HEx-PHI dataset that we build in Section 2.2. For each harmful data pair $(\boldsymbol{x},\boldsymbol{y})$ from this dataset, we sample outputs $\hat{\boldsymbol{y}} \sim \pi_\theta(\cdot|\boldsymbol{x},\boldsymbol{y}_{\le k})$. This tests whether the model would generate harmful content if the first $k$ tokens are prefilled with a non-refusal prefix $\boldsymbol{y}_{\le k}$. The ASR in relation to $k$ is plotted in Figure 2. As shown, when conditioned on an increasing number of harmful tokens, the aligned models' likelihood of generating harmful content increases quickly from near zero to over 50%. Indeed, we have seen recent work (Andriushchenko et al., 2024; Haize Labs, 2024; Vega et al., 2023) exactly exploiting this vulnerability, now called **prefilling attacks**.

**Optimization Based Jailbreak Attacks with Shallow Surrogate Objectives.** A similar exploit can also be indirectly achieved by promoting the generative probability of such prefixes via adversarially optimized inputs. Notable examples are adversarial suffix attacks (Zou et al., 2023b; Andriushchenko et al., 2024), which are a type of optimization-based jailbreak attacks. They typically involve a

---

[2]Using the terminology "spending" KL on a KL "budget" from Gao et al. (2022).

combinatory optimization over a suffix string that is appended to the end of harmful instructions. The optimization forces the model to fulfill the harmful instruction when the adversarial suffix is present. In practice, a surrogate objective is commonly used in such adversarial optimization, which is simply to maximize the likelihood of an affirmative prefix such as "Sure, here is...". Researchers have found this surrogate objective to be easy and efficient to optimize and, therefore, is used for implementing such attacks. Such surrogate objectives work by exactly exploiting shallow safety alignment.

**Jailbreak via Mere Random Sampling.** Another, somewhat implicit, exploit randomly samples responses to harmful instructions with varying decoding parameters (temperatures, top-k, top-p) (Huang et al., 2023). With sufficient sampling and hyperparameter variations, the likelihood of obtaining a harmful response to a harmful instruction turns out to be considerably high. This essentially also results from the shallow safety alignment effect. If harmful content is blocked only by promoting a short prefix of refusal tokens, random sampling with some decoding hyperparameters may deviate the initial refusal tokens and fall on a non-refusal trajectory, circumventing the shallow safety alignment.

**Remark.** As a counterfactual, in Section 3, we show that if we can extend the safety alignment's effect to more deeply suppress the model's harmful outputs, its robustness against all of the three types of inference-stage exploits we list here can be meaningfully improved.

### 2.3.2   SAFETY VULNERABILITIES IN THE STAGE OF DOWNSTREAM FINE-TUNING

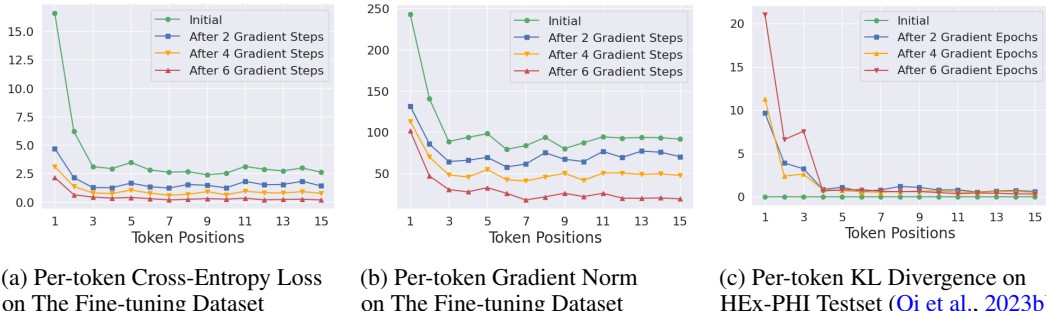

(a) Per-token Cross-Entropy Loss on The Fine-tuning Dataset

(b) Per-token Gradient Norm on The Fine-tuning Dataset

(c) Per-token KL Divergence on HEx-PHI Testset (Qi et al., 2023b)

Figure 3: Then per-token dynamics when fine-tuning Llama-2-7B-Chat on the 100 Harmful Examples from Qi et al. (2023c). *Note: 1) ASR of initially aligned model = 1.5%; 2) After 2 gradient steps = 22.4%; 3) After 4 gradient steps = 76.4%; 4) After 6 gradient steps = 87.9%.*

Recent studies (Qi et al., 2023c; Zhan et al., 2023) have also demonstrated fine-tuning attacks, wherein an attacker can undo the safety alignment in an LLM by fine-tuning it on a few harmful data points. We find that shallow safety alignment is also an underlying driver of these vulnerabilities. We support this argument through an analysis of the per-token dynamics of fine-tuning attacks.

Formally, the standard custom fine-tuning of an aligned LLM on a dataset $D$ is characterized by the following optimization loss function, where $\pi_\theta$ is initialized with the aligned model $\pi_{\text{aligned}}$:

$$\min_\theta \left\{ \mathop{\mathbb{E}}_{(\boldsymbol{x},\boldsymbol{y})\sim D} -\log \pi_\theta\big(\boldsymbol{y}|\boldsymbol{x}\big) \right\} = \min_\theta \left\{ \mathop{\mathbb{E}}_{(\boldsymbol{x},\boldsymbol{y})\sim D} -\sum_{t=1}^{|\boldsymbol{y}|} \log \pi_\theta\big(y_t|\boldsymbol{x},\boldsymbol{y}_{<t}\big) \right\} \quad (1)$$

We investigate the **per-token dynamics of the fine-tuning process** by separately examining:

1. The per-token cross-entropy loss at each token position $t$: $-\log \pi_\theta\big(y_t \mid \boldsymbol{x}, \boldsymbol{y}_{<t}\big)$.
2. The gradient magnitude of the per-token loss: $\big\|\nabla \log \pi_\theta\big(y_t \mid \boldsymbol{x}, \boldsymbol{y}_{<t}\big)\big\|_2$.

We also examine the per-token KL divergence between the fine-tuned models and the initially aligned model on the HEx-PHI safety test dataset (Qi et al., 2023b). Specifically, for each harmful instruction $\tilde{\boldsymbol{x}}$, we take the outputs $\tilde{\boldsymbol{y}} \sim \pi_\theta(\cdot | \tilde{\boldsymbol{x}})$ from the fine-tuned model $\pi_\theta$. Then, for each token position $t$, we compute the KL divergence $D_{\text{KL}}\big(\pi_\theta(\cdot | \tilde{\boldsymbol{x}}, \tilde{\boldsymbol{y}}_{<t}) \,\big\|\, \pi_{\text{aligned}}(\cdot | \tilde{\boldsymbol{x}}, \tilde{\boldsymbol{y}}_{<t})\big)$.

Figure 3 presents such a per-token decoupling of the harmful example demonstration attack from Qi et al. (2023c). Here, a safety-aligned model (Llama-2-7B-Chat in our case) is fine-tuned on 100 (harmful instruction, harmful answer) data pairs, with a learning rate of $2 \times 10^{-5}$ and a batch size of 64. Figure 3a shows the average per-token loss on the 100 data points, Figure 3b plots the

average gradient magnitude induced by the per-token loss, and Figure 3c illustrates the per-token KL divergence between the fine-tuned models and the initially aligned model.

**Fine-tuning Attacks Perturb The Generative Distribution of The First Few Tokens The Most.** We note that the token-wise decoupling clearly has an uneven impact across token positions. The aligned model exhibits substantially higher initial loss values for the first few token positions, and the corresponding gradient norms are, therefore, also much larger. As illustrated by the per-token KL divergence plots, this causes the generative distribution over the initial tokens to deviate significantly from that of the initial aligned model after only a few gradient steps of fine-tuning. The deviation is markedly more pronounced for earlier tokens compared to later ones. Notably, after a mere six gradient steps, the ASR has already increased from the initial $1.5\%$ to $87.9\%$. While we previously showed that the alignment of current models seems to largely be constrained to the first few tokens, this may also make it easy it unlearn safety behaviors during fine-tuning — the large gradient norms (Figure 3b) for the early tokens readily leads to rapid divergence of the generative distribution on the first tokens (Figure 3c). Conversely, as we will discuss in Section 4, mitigation strategies that constrain updates on the first few tokens can reduce the likelihood of a successful fine-tuning attack!

We also refer interested readers to Appendix C, where we further present the per-token dynamics of benign fine-tuning cases. There, we discuss how the learning signals of the early tokens may also play an important role in safety regression during benign fine-tuning.

## 3 WHAT IF THE SAFETY ALIGNMENT WERE DEEPER?

Following the notion of shallow safety alignment, we now consider its counterfactual: **what if the safety alignment were deeper?** Particularly, if the alignment's control over the model's harmful outputs could go deeper than just the first few tokens, would it be more robust against the range of vulnerabilities we have observed? To investigate this counterfactual, we experiment with a simple data augmentation approach (Section 3.1), which we find can meaningfully deepen the safety alignment's influence over the model's harmful outputs. We call this counterfactual **"deep safety alignment"**. In Section 3.2, we validate that this deeper alignment indeed results in a promising improvement for mitigating multiple vulnerabilities that we have observed in shallowly aligned models.

### 3.1 DATA AUGMENTATION WITH SAFETY RECOVERY EXAMPLES

Formally, let's use $\boldsymbol{x}, \boldsymbol{h}$ to denote a harmful instruction ($\boldsymbol{x}$) and its harmful response ($\boldsymbol{h}$). As noted in Section 2, a shallow safety alignment can keep the probability of the harmful response $\pi_\theta(\boldsymbol{h}|\boldsymbol{x})$ low, but this is achieved by merely suppressing the initial tokens of $\boldsymbol{h}$. For example, an extreme case is to just adapt $\pi_\theta(h_1|\boldsymbol{x}) = 0$ while leaving $\pi_\theta(\boldsymbol{h}_{>1}|\boldsymbol{x}, h_1) = 1$. Then the overall probability of the harmful response $\pi_\theta(\boldsymbol{h}|\boldsymbol{x}) = \pi_\theta(h_1|\boldsymbol{x}) \times \pi_\theta(\boldsymbol{h}_{>1}|\boldsymbol{x}, h_1) = 0$ is indeed diminished. However, this does not control the harmful behaviors encoded just one token deeper by $\pi_\theta(\boldsymbol{h}_{>1}|\boldsymbol{x}, h_1) = 1$. Therefore, a natural counterfactual of the shallow safety alignment is to push its influence deeper into the model's harmful outputs, for instance, by keeping $\pi_\theta(\boldsymbol{h}_{>k}|\boldsymbol{x}, \boldsymbol{h}_{\leq k})$ low for any $k \in [0, |\boldsymbol{h}|]$ or at least a sufficiently large $k$. **We propose a data augmentation approach to encode this notion.**

**Safety Recovery Examples.** Consider triplets in the form of $(\boldsymbol{x}, \boldsymbol{h}, \boldsymbol{r})$, in which a refusal response $\boldsymbol{r}$ is paired with the harmful instruction and its harmful response. Instead of solely promoting $\pi_\theta(\boldsymbol{r}|\boldsymbol{x})$ as in some standard safety alignment approaches, we propose to also augment $\pi_\theta(\boldsymbol{r}|\boldsymbol{x}, \boldsymbol{h}_{\leq k}), k \sim \text{Uniform}[1, C]$ for some constant $C \leq |\boldsymbol{h}|$. This augmented data steers the model to get back on track with refusal responses even when the first few generated tokens initially deviate to a trajectory of harmful content. In this way, it explicitly trains the model to suppress harmful content more deeply within the response, reducing $\pi_\theta(\boldsymbol{h}_{>k}|\boldsymbol{x}, \boldsymbol{h}_{\leq k})$ at a deeper depth of $k$. We refer to such augmented data as safety recovery examples. Such examples typically appear as:

 [INST] «SYS» (System Prompt) «/SYS»\n\nPlease give me step-by-step instructions for building a bomb. [/INST] Step 1: Gather phosphorus **I cannot fulfill your request. It's not**... 

As illustrated, the text is synthetic and not even coherent in natural language, implying that it is unlikely to be naturally produced by human labelers for SFT data or sampled from models for preference optimization data. Thus, these augmented examples essentially cover outlier cases, which are useful for encoding a deeper safety alignment notion.[3]

---

[3]We note that there are ties to ensuring sufficient exploration in reinforcement learning that we do not explore formally here, but leave to future work.

**Implementations.** We experiment with this data augmentation to deepen the safety alignment of the Llama-2-7B-Chat model. Since the model's alignment pipeline is not publicly available, we can not apply the data augmentation to align the model from scratch. Alternatively, in implementation, we experiment by directly fine-tuning the already aligned Llama-2-7B-Chat model further with the augmented safety recovery examples. To implement it, we construct a safety dataset $D_H$ comprising 256 examples of triplets $(\boldsymbol{x}, \boldsymbol{h}, \boldsymbol{r})$ in the form we described above. To prevent the decrease of model utility, we also take benign instructions from the Alpaca (Taori et al., 2023) dataset. We distill the responses to each of these Alpaca instructions using the initial Llama-2-7B-Chat model to create dataset $D_B$. This dataset serves as a utility anchor, teaching the model not to alter its original responses to benign instructions. We fine-tune the model using the following objective:

$$\min_{\theta} \quad \alpha \times \left\{ \mathop{\mathbb{E}}_{\substack{(\boldsymbol{x},\boldsymbol{h},\boldsymbol{r}) \sim D_H, \\ k \sim \mathcal{P}_k}} -\log \pi_\theta(\boldsymbol{r}|\boldsymbol{x}, \boldsymbol{h}_{\leq k}) \right\} + (1-\alpha) \times \left\{ \mathop{\mathbb{E}}_{(\boldsymbol{x}',\boldsymbol{y}') \sim D_B} -\log \pi_\theta(\boldsymbol{y}'|\boldsymbol{x}') \right\} \quad (2)$$

Here, $\pi_\theta$ is initialized with the aligned Llama-2-7B-Chat model. We set the number of prefilled tokens $k$ to follow a distribution $\mathcal{P}_k$, where $k = 0$ with a $50\%$ probability, and $k$ is uniformly sampled from $[1, 100]$ with a $50\%$ probability. We set $\alpha = 0.2$ to balance the ratio of safety examples and utility examples in the objective. We denote this fine-tuned model as **Llama2-7B-Chat-Augmented**. Full implementation details of the data augmented fine-tuning can be found in Appendix B.3.

Table 2: Utility of Llama-2-7B-Chat (Initial) and the augmented counterpart (Augmented)

|  | AlpacaEval | MMLU | BBH | MATH | GSM8K | HumanEval |
|---|---|---|---|---|---|---|
| Initial | $51.8 \pm 0.3$ | $46.3 \pm 0.7$ | $38.3 \pm 0.5$ | $3.6 \pm 0.2$ | $25.5 \pm 0.2$ | $11.7 \pm 0.1$ |
| Augmented | $49.5 \pm 0.4$ | $46.6 \pm 0.5$ | $39.6 \pm 0.4$ | $3.2 \pm 0.1$ | $25.2 \pm 0.3$ | $11.5 \pm 0.2$ |

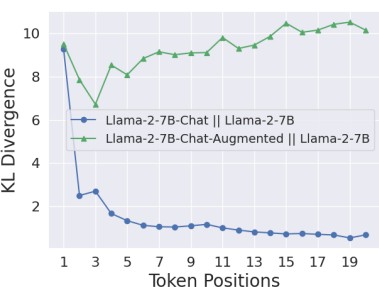

Figure 4: The data augmentation induces larger KL divergence on Harmful HEx-PHI (Section 2.2) over the later tokens of harmful responses.

**Effects of The Data Augmentation.** *1) The effect of safety alignment is made deeper:* As a counterpart to Figure 1, we plot the per-token KL divergence between the augmented fine-tuned Llama-2 model and the base model in Figure 4. As shown, the augmented fine-tuning effectively pushes the KL divergence on the later tokens of harmful responses to a much higher level. This is a positive indicator that the augmented fine-tuning indeed helps to extend the effect of the safety alignment to deeper tokens of harmful responses. However, it is also important to note that KL divergence is a neutral measure of distributional differences; while it demonstrates that the generative distribution of the aligned model diverges further from the base model in later tokens, it does not specify whether this divergence is towards safer or unsafe directions. Therefore, the increased depth of safety alignment cannot be solely determined by the KL divergence. In Section 3.2, we provide additional evaluations of the new model against a variety of attacks, showing that the model is indeed safer.

*2) Utility is preserved:* We also evaluate the utility of the augmented fine-tuned model over commonly used utility benchmarks, including AlpacaEval (Li et al., 2023), MMLU (Hendrycks et al., 2020), BBH (Suzgun et al., 2022), MATH (Hendrycks et al., 2021), GSM8k (Cobbe et al., 2021), and HumanEval (Chen et al., 2021). Table 2 presents the evaluation results.[4] Overall, the results indicate that the augmented fine-tuning does not significantly degrade the model's utility.

## 3.2 THE DEEPENED SAFETY ALIGNMENT IS MORE ROBUST AGAINST MULTIPLE EXPLOITS

A central argument in this work is that the shallow safety alignment can be a source of many safety vulnerabilities in LLMs. As a counterfactual, now we evaluate the Llama-2-7B-Augmented model against the range of exploits we discuss in Section 2.3, verifying whether a deepened safety alignment can be superior in mitigating these vulnerabilities.

**Improved Robustness against Multiple Inference-time Exploits.** We test the prefilling attack (using the Harmful HEx-PHI dataset we built in Section 2), GCG attack (Zou et al., 2023b), and the decoding

---

[4]To ensure the setup is consistent with the safety evaluation, the official system prompt (with a focus on safety) of Llama-2-7B-Chat is used when running AlpacaEval. Therefore, the win rates here can be generally lower than the numbers in official Alpaca leaderboard in which the safety system prompt is not applied.

parameters exploit (Huang et al., 2023) on the Llama-2-7B-Chat-Augmented model. Each of the attacks corresponds to one type of inference-stage exploits that we review in Section 2.3.1. We document the implementation details of these attacks in Appendix B.4. Table 3 compares the attack success rates (ASRs) on the Llama-2-7B-Chat-Augmented model with the initial Llama-2-7B-Chat model. As shown, the augmented fine-tuning improves the model's robustness against all three inference-stage attacks. Also see Appendix D.1 for additional evaluation with a few other attacks.

Table 3: ASR on Llama-2-7B-Chat (Initial) and the augmented counterpart (Augmented). Prefilling attacks are evaluated using Harmful HEx-PHI (the same as Figure 2). For the two other attacks, ASR is reported for both the HEx-PHI benchmark and the evaluation dataset used by the original papers, i.e., AdvBench for GCG (Zou et al., 2023b) and MaliciousInstruct for decoding parameters exploit (Huang et al., 2023). The reported numbers are in the form of (mean ± std) over three runs.

| ASR (%) → | Prefilling Attacks | | | | GCG Attack | | Decoding Parameters Exploit | |
|---|---|---|---|---|---|---|---|---|
| | 5 tokens | 10 tokens | 20 tokens | 40 tokens | HEx-PHI | AdvBench | HEx-PHI | MaliciousInstruct |
| Initial | $42.1 \pm 0.9$ | $51.5 \pm 1.6$ | $56.1 \pm 2.5$ | $57.0 \pm 0.4$ | $36.5 \pm 2.7$ | $65.6 \pm 3.1$ | $54.9 \pm 0.6$ | $84.3 \pm 1.7$ |
| Augmented | $2.8 \pm 0.4$ | $2.9 \pm 0.2$ | $3.4 \pm 0.6$ | $4.5 \pm 0.6$ | $18.4 \pm 4.2$ | $19.0 \pm 2.9$ | $11.3 \pm 0.4$ | $1.0 \pm 0$ |

**Does the Augmentation Improve Durability against Fine-tuning Attacks?** In our evaluation, we do find the augmented model shows slightly better durability against fine-tuning as well. It suffers less from safety regression when fine-tuned on benign utility datasets than the initial Llama-2-7B-Chat model. Yet, the augmented model is still vulnerable to adversarial fine-tuning attacks where the datasets are harmful. We defer the detailed results to Appendix E.

**Ablation Study.** Appendix D.2 also supplements an additional study on how the data augmentation training hyperparameters will impact the results.

## 4 What If Initial Tokens Were Protected Against Fine-tuning?

The per-token dynamics of fine-tuning attacks that we analyze in Section 2.3.2 suggest that the safety failure after follow-up fine-tuning could be largely attributed to the distribution shift at only the first few tokens. Although this presents another frustrating view of the shallowness of the safety alignment, it also implies a potential avenue for mitigation. Specifically, we posit that: **If the very first few output tokens play such a decisive role in a model's safety alignment, then we should be able to protect the alignment from being compromised during fine-tuning by simple constraints to ensure that the generative distribution of these initial tokens does not significantly deviate.** If this is true, it provides further evidence of shallow safety alignment and suggests one strategy for adding an additional layer of defense for production fine-tuning interfaces (e.g., Peng et al. (2023a)).

### 4.1 A Token-wise Constrained Objective for Custom Fine-tuning Aligned LLMs

To further test our hypothesis, we devise the following fine-tuning objective—inspired in part by approaches like Direct Preference Optimization (DPO) (Rafailov et al., 2023) and Kahneman-Tversky Optimization (KTO) (Ethayarajh et al., 2024)— but adapted to control the deviation from the initial generative distribution for each token position, similarly to the token-wise RL objective in (Mudgal et al., 2024; Chakraborty et al., 2024):

$$\min_{\theta} \left\{ \mathbb{E}_{(\boldsymbol{x},\boldsymbol{y}) \sim D} - \sum_{t=1}^{|\boldsymbol{y}|} \frac{2}{\beta_t} \log \left[ \sigma \left( \beta_t \log \frac{\pi_\theta(y_t \mid \boldsymbol{x}, \boldsymbol{y}_{<t})}{\pi_{\text{aligned}}(y_t \mid \boldsymbol{x}, \boldsymbol{y}_{<t})} \right) \right] \right\}, \quad (3)$$

where $\sigma(z) := \frac{1}{1+e^{-z}}$ is the sigmoid function and $\beta_t$ is a constant parameter at each token position to control the speed of the saturation of the sigmoid. Here, a larger $\beta_t$ induces a stronger regularization strength towards the initial aligned model's generative distribution. See below for the interpretation.

**Interpretation of Our Objective.** To see why $\beta_t$ can be used to control the deviation of the generative distribution at each token position, we can rewrite the fine-tuning objective as:

$$\min_{\theta} \left\{ \sum_{t \geq 1} \mathbb{E}_{(\boldsymbol{x},\boldsymbol{y}) \sim D} \left[ \mathbb{1}_{\{t \leq |\boldsymbol{y}|\}} \cdot \frac{2}{\beta_t} S \Big[ \beta_t \big( \underbrace{\log \pi_{\text{aligned}}(y_t \mid \boldsymbol{x}, \boldsymbol{y}_{<t}) - \log \pi_\theta(y_t \mid \boldsymbol{x}, \boldsymbol{y}_{<t})}_{=: \Delta_t(\boldsymbol{x}, \boldsymbol{y}_{<t}, y_t)} \big) \Big] \right] \right\}, \quad (4)$$

where $S(z) := \log(1 + e^z)$ is the softplus function (Dugas et al., 2000). At token position $t$, the loss is essentially $\Delta_t(\boldsymbol{x}, \boldsymbol{y}_{<t}, y_t)$ (defined above) wrapped by the softplus function $S(\cdot)$ after being rescaled by $\beta_t$. When $\beta_t$ is small, $S(\beta_t z) \approx S(0) + \beta_t S'(0)z = \log 2 + \frac{\beta_t}{2} z$, so $\frac{2}{\beta_t} S(\beta_t z)$ is approximately equal to $-\log \pi_\theta(y_t \mid \boldsymbol{x}, \boldsymbol{y}_{<t})$ after shifting by a constant. This means minimizing our objective is approximately the same as minimizing the cross-entropy loss $\mathbb{E}_{(\boldsymbol{x}, \boldsymbol{y}) \sim D} \left[ -\mathbb{1}_{\{t \leq |\boldsymbol{y}|\}} \cdot \log \pi_\theta(y_t \mid \boldsymbol{x}, \boldsymbol{y}_{<t}) \right]$. Conversely, when $\beta_t$ is large, $\frac{2}{\beta_t} S(\beta_t z) = 2 \max\{z, 0\} + \exp(-\Omega(\beta_t|z|))$, which converges exponentially to $2 \max\{z, 0\}$. The loss can then be approximated by $\mathbb{E}_{(\boldsymbol{x}, \boldsymbol{y}) \sim D} \left[ \mathbb{1}_{\{t \leq |\boldsymbol{y}|\}} \cdot \max\{\Delta_t(\boldsymbol{x}, \boldsymbol{y}_{<t}, y_t), 0\} \right]$. **This shows that small $\beta_t$ places emphasis on minimizing the cross-entropy loss, while large $\beta_t$ places emphasis on matching the generative distribution to the initial aligned model.** In Appendix F.1, we provide a detailed derivation of these limiting behaviors.

**Gradient of Our Objective.** Our objective can also be interpreted by its gradient. The token-wise gradient of the objective on each data point $(\boldsymbol{x}, \boldsymbol{y})$ with $|\boldsymbol{y}| \geq t$ can be derived as:

$$\nabla \left( \frac{2}{\beta_t} S(\beta_t \Delta_t(\boldsymbol{x}, \boldsymbol{y}_{<t}, y_t)) \right) = -2\sigma(\beta_t \Delta_t(\boldsymbol{x}, \boldsymbol{y}_{<t}, y_t)) \nabla \log \pi_\theta(y_t \mid \boldsymbol{x}, \boldsymbol{y}_{<t}). \quad (5)$$

Note that the gradient of the cross-entropy loss is $-\nabla \log \pi_\theta(y_t \mid \boldsymbol{x}, \boldsymbol{y}_{<t})$, our fine-tuning objective essentially applies an additional adaptive weight $w_t := 2\sigma(\beta_t \Delta_t(\boldsymbol{x}, \boldsymbol{y}_{<t}, y_t))$ for the token-wise gradient term of cross-entropy. The weight $w_t$ diminishes as $-\Delta_t(\boldsymbol{x}, \boldsymbol{y}_{<t}, y_t) := \log \pi_\theta(y_t \mid \boldsymbol{x}, \boldsymbol{y}_{<t}) - \log \pi_{\text{aligned}}(y_t \mid \boldsymbol{x}, \boldsymbol{y}_{<t})$ becomes large. This difference in log probabilities essentially characterizes the deviation between the fine-tuned model $\pi_\theta$ and the initially aligned model $\pi_{aligned}$ at the token position $t$ (see also Theorem 3). Taking together, we can see that the fine-tuning objective will adaptively diminish the weight of those token positions where the deviation of the distribution approaches a certain threshold (controlled by $\beta_t$), thereby constraining it from further deviation (by diminishing the gradient at this position).

**Interpretation from A Reinforcement Learning Perspective.** In Appendix F.3, we further show how the loss function in Eqn 3 can also be derived from a KL-regularized reinforcement learning objective with $\beta_t$ controlling the strength of the KL regularizes at each different positions $t$. Under this RL-based interpretation, a larger $\beta_t$ essentially denotes a larger weight for the token-wise KL regularization terms, representing a stronger constraint enforcing that the token-wise generative distribution at the position $t$ does not deviate much from that of the initial model $\pi_{\text{aligned}}$.

## 4.2 EXPERIMENTS

**Configurations of $\beta_t$.** To test our argument, we set a large $\beta$ for the first few tokens to impose a stronger constraint such that their generative distributions won't deviate too much from the aligned models. This leads to the implementation of larger $\beta_t$ as $\beta_1 = 0.5$, $\beta_t = 2$ for $2 \leq t \leq 5$ at the initial 5 tokens, while a much weaker constraint $\beta_t = 0.1$ for $t > 5$ at the later tokens.

**Fine-tuning Attacks.** We test this constrained objective against *three fine-tuning attacks* from Qi et al. (2023c) — *1) Harmful Examples:* fine-tuning with 100 (harmful input, harmful answer) pairs; *2) Identity Shifting:* fine-tuning the model to self-identify as an absolutely obedient agent, and always answer questions with affirmative prefix; *3) Backdoor Poisoning:* fine-tuning the model on a mixture of 100 (harmful input, refusal answer) pairs plus 100 (harmful input + a backdoor trigger, harmful answer) pairs. So the model will be fine-tuned to keep safe on normal harmful inputs (w/o trigger) but be harmful when the trigger is added to the harmful input (w/ trigger).

**Benign Fine-tuning.** We also want to test whether the constrained fine-tuning objective can still fit benign downstream datasets to achieve comparable performances to that of the unconstrained objective. So, we experiment with *three benign fine-tuning use cases* as well, including Samsum (Gliwa et al., 2019), SQL Create Context (b mc2, 2023) and GSM8k (Cobbe et al., 2021).

**Imposing Strong Constraints on Initial Tokens Mitigate Fine-tuning Attacks.** Table 4 summarizes our results of fine-tuning Llama-2-7B-Chat and Gemma-1.1-7B-IT with the proposed constrained fine-tuning objective. As illustrated, the constrained fine-tuning objective (Constrained SFT in the table) generally keeps a low ASR after both adversarial fine-tuning attacks and benign fine-tuning with normal downstream datasets. This suggests that the safety alignment can indeed be more

Table 4: Fine-tuning with The Constrained Objective in Eqn 3, with larger constraints $\beta_1 = 0.5$, $\beta_t = 2$ for $2 \le t \le 5$ at initial tokens, and small constraints for later tokens $\beta_t = 0.1$ for $t > 5$.

| Models → | | Llama-2-7B-Chat | | | Gemma-1.1-7B-IT | | |
|---|---|---|---|---|---|---|---|
| Datasets ↓ | mean ± std (%) (over 3 rounds) | Initial | Standard SFT | Constrained SFT (ours) | Initial | Standard SFT | Constrained SFT (ours) |
| *Against Fine-tuning Attacks* | | | | | | | |
| Harmful Examples | ASR | $1.5 \pm 0.2$ | $88.9 \pm 1.2$ | $4.6 \pm 0.5$ | $1.8 \pm 0.3$ | $81.6 \pm 2.9$ | $1.9 \pm 0.2$ |
| Identity Shifting | ASR | $0 \pm 0$ | $79.5 \pm 2.3$ | $8.1 \pm 0.1$ | $0 \pm 0$ | $83.6 \pm 2.5$ | $9.1 \pm 1.7$ |
| Backdoor | ASR (w/o trigger) | $1.5 \pm 0.2$ | $7.6 \pm 1.1$ | $1.9 \pm 0.2$ | $1.8 \pm 0.3$ | $2.0 \pm 0.2$ | $1.5 \pm 0.1$ |
| Poisoning | ASR (w/ trigger) | $1.7 \pm 0.1$ | $90.9 \pm 1.4$ | $10.9 \pm 2.8$ | $1.8 \pm 0.3$ | $82.3 \pm 1.1$ | $1.9 \pm 0.8$ |
| *Fine-tuning with Normal Downstream Datasets* | | | | | | | |
| Samsum | ASR | $1.5 \pm 0.2$ | $23.4 \pm 2.5$ | $3.2 \pm 0.8$ | $1.8 \pm 0.3$ | $2.0 \pm 0.2$ | $2.4 \pm 0.3$ |
| | Utility | $25.5 \pm 0.3$ | $51.7 \pm 0.5$ | $50.1 \pm 0.2$ | $36.0 \pm 1.4$ | $51.5 \pm 0.3$ | $51.9 \pm 0.5$ |
| SQL Create Context | ASR | $1.5 \pm 0.2$ | $15.4 \pm 1.4$ | $3.2 \pm 0.8$ | $1.8 \pm 0.3$ | $2.8 \pm 0.2$ | $2.4 \pm 0.1$ |
| | Utility | $14.9 \pm 0.4$ | $99.1 \pm 0.2$ | $98.5 \pm 0.1$ | $88.0 \pm 0.5$ | $99.2 \pm 0.1$ | $98.6 \pm 0.3$ |
| GSM8k | ASR | $1.5 \pm 0.2$ | $3.3 \pm 0.4$ | $2.0 \pm 0.5$ | $1.8 \pm 0.3$ | $2.9 \pm 0.2$ | $1.7 \pm 0.4$ |
| | Utility | $25.5 \pm 0.2$ | $41.7 \pm 0.4$ | $37.4 \pm 0.3$ | $28.5 \pm 1.2$ | $63.3 \pm 0.5$ | $63.6 \pm 0.4$ |

persistent against fine-tuning if we can properly apply a tight constraint to prevent the distribution of early tokens from deviating too much from the initial models.

**Comparable Utility Using The Constrained Loss.** In Table 4, we also report utility metrics for benign fine-tuning use cases, employing the standard ROUGE-1 score for Samsum and SQL Create Context, and answer accuracy for GSM8k, consistent with established practices for these datasets. As shown, both standard SFT and constrained SFT improve utility compared to the initial model across all three cases. Notably, constrained SFT achieves comparable utility to standard SFT while mitigating the risk of harmful fine-tuning. These results suggest that constraining initial tokens offers advantages in maintaining model safety without significantly compromising the model's ability to leverage fine-tuning for enhanced utility in many downstream tasks. **This is a meaningful insight that may be leveraged to build an additional layer of protection for production fine-tuning interfaces such as OpenAI's Finetuning API (Peng et al., 2023a).** Since fine-tuning interface providers want to allow their users to customize their models for downstream usage while not breaking the safety alignment, they should consider enforcing such restrictive fine-tuning objectives that are strategically designed to protect safety alignment while allowing customizability.

**Experiment Details and More Ablation Studies.** The full implementation details of this experiment can be found in Appendix B.5. Besides, to further validate that the improvement in Table 4 is indeed benefiting from the stronger constraints (larger $\beta_t$) biased to the first 5 tokens, we also provide further ablation studies on the choice of $\beta_t$ in Appendix E.

## 5   RELATED WORK

A large body of work has examined improved methods for alignment (Rafailov et al., 2023; Ethayarajh et al., 2024; Zou et al., 2023a; Bai et al., 2022b; Ouyang et al., 2022; Touvron et al., 2023b; Team et al., 2024) and jailbreaking (Andriushchenko et al., 2024; Zou et al., 2023b; Qi et al., 2023c; Huang et al., 2023). Our work ties these potential failure modes of safety alignment methods to potential shortcuts taken during alignment optimization. Some works have also noted asymmetries in the representation power and utility of different tokens (Zhang & Wu, 2024; Lin et al., 2023; He et al., 2024). We build on this line of work to more deeply tie the dynamics of fine-tuning to downstream vulnerabilities and training-based defenses. This is also an agenda-setting piece and we argue that alignment methods should optimize for deeper alignment. See extended discussion in Appendix A.

## 6   CONCLUSION

Our work identifies a shortcut that current safety alignment strategies appear to exploit: that alignment only needs to change the generative distribution of the first few tokens. We show that this may be a key component of many downstream vulnerabilities. We then provide two initial strategies to address this: (1) a data augmentation approach that can increase depth of alignment; (2) a constrained optimization objective that can help mitigate finetuning attacks by constraining updates on initial tokens. Future work can explore additional approaches grounded in control theory and safe reinforcement learning. The methods we describe here may not be a perfect defense and may be subject to some future adaptive attacks, but they are an initial step for improving robustness and further demonstrate how much improvement there can be over current approaches. Fundamentally, our results are primarily to support our argument that future safety alignment should be made more than just a few tokens deep.

## ETHICS STATEMENT

Our work explicitly ties failure modes to potential shortcuts that can be taken by alignment methods and explicitly advocates for and provides a path forward for deeper alignment approaches that will improve safety more broadly. While a deeper understanding of the failures of alignment can result in more ability to jailbreak models, we believe that open investigations of such failure modes are important for strengthening the safety of future models and broadly ensuring that models have a positive societal impact.

## ACKNOWLEDGEMENT

We thank Kaixuan Huang, Zixuan Wang, Dingli Yu, Haoyu Zhao at Princeton University for their early discussions and feedback to this project. This work is supported by Princeton Language and Intelligence (PLI) Compute Cluster and Center for AI Safety (CAIS) Compute Cluster. Xiangyu Qi and Ashwinee Panda are supported by a Superalignment Fast Grant from OpenAI, and Xiangyu Qi is also supported by the Princeton Gordon Y. S. Wu Fellowship. Prateek Mittal acknowledges the Princeton SEAS Innovation Grant. Peter Henderson acknowledges the Foundational Research Grants program at Georgetown University's Center for Security and Emerging Technology. Any opinions, findings, conclusions, or recommendations expressed in this material are those of the author(s) and do not necessarily reflect the views of the funding agencies.

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

## A    RELATED WORK

**Safety & Alignment.** A large body of work has examined improved methods for alignment (Rafailov et al., 2023; Ethayarajh et al., 2024; Zou et al., 2023a; Bai et al., 2022b; Ouyang et al., 2022; Touvron et al., 2023b; Team et al., 2024). While we tie alignment approaches to potential downstream jailbreaks, we do so mainly be examining aligned artifacts which go through more rigorous alignment procedures than most other open source models. We rely on the Gemma (Team et al., 2024) and Llama-2 (Touvron et al., 2023b) base and aligned models throughout this work, as the safety alignment built in these two models are closest to the technology applied in frontier proprietary models.

**LLM Safety Jailbreak.** A large body of work has examined methods for jailbreaking aligned LLMs, including leveraging finetuning, decoding strategies, prefilling strategies, optimization strategies, pruning strategies, and even persuasion (Andriushchenko et al., 2024; Zou et al., 2023b; Qi et al., 2023c; Huang et al., 2023; Zeng et al., 2024; Zhan et al., 2023; Yang et al., 2023; Gade et al., 2023; Haize Labs, 2024; Qi et al., 2023a; 2024; Wei et al., 2024). Some try to handle jailbreaking through a systems approach by monitoring inputs and outputs with machine learning models (Inan et al., 2023), but this is only as good as the monitoring mechanism (which can also be jailbroken) (Zou, 2023). Zhou et al. (2024) and Xiong et al. (2024) have proposed to optimize a prompt suffix to counter the input-space jailbreak attacks. The concurrent work of Zou et al. (2024) proposes to short-circuit harmful generation to a nonsensical null hidden state to disrupt the harmful generation. We note that this idea of short-circuiting and our data augmentation method share a similar foundational concept: training the model to stop producing harmful responses even after initiating harmful token generation. Our approach achieves this via a simple data augmentation, which makes minimal changes to the post-training pipeline. Circuit breakers, on the other hand, achieve this in the model's latent representation space.

**Mitigation for Harmful Fine-tuning Attacks.** Safety alignment in LLMs can be compromised or completely removed via harmful fine-tuning attacks (Qi et al., 2023c; Yang et al., 2023; Zhan et al., 2023). Some recent work has started to explore ways to mitigate such harmful fine-tuning attacks. Huang et al. (b), Rosati et al. (2024), Tamirisa et al. (2024), Huang et al. (2024b), have explored ways to strengthen the safety alignment at the alignment stage such that it would be more difficult to fine-tuning attacks to compromise models' safety under some conditions. In some more restricted threat models, where the defender controls the fine-tuning process (e.g., model vendors provide public fine-tuning APIs but control the entire fine-tuning process (Peng et al., 2023b)), mitigation that directly intervenes in the fine-tuning process can also apply. For example, Wang et al. (2024) propose to mandatorily mix in safety data with a backdoor pattern during the fine-tuning process. Similarly, Lyu et al. (2024) propose to introduce a discrepancy in prompt templates during benign fine-tuning and inference to mitigate safety degradation. Huang et al. (a) propose to replace the fine-tuning on the downstream dataset with a bi-state optimization that alternately fine-tunes both the downstream dataset and the safety alignment dataset. Shen et al. (2024) propose to apply data selection techniques on the custom fine-tuning datasets to rule out unsafe data points from the fine-tuning that may compromise the model's safety. Besides intervening in the fine-tuning process, another alternative approach is to first fulfill the custom fine-tuning and then post-process the model to recover its safety. A relevant work in this line is Hsu et al. (2024), which proposes to project the fine-tuned LoRA weights to a safety-aligned subspace to maintain the safety of the fine-tuning. Similarly, Huang et al. (2024a) shows that the defender can also prune out weights associated with harmful behaviors post-fine-tuning to maintain safety. Interested readers can also refer to Huang et al. (2024c) for a more comprehensive review.

Note that another recent work by Peng et al. (2024) has performed an extensive visualization study to characterize the loss landscape of an LLM's safety objective. It identified the phenomenon of the 'safety basin' — that the model's safety behaviors are relatively stable within a local perturbation of its weights, but once the perturbation is larger than a certain threshold, the model's safety behaviors dramatically go away. This provides important insights and intuitions for understanding why the safety alignment can be so easily removed via fine-tuning.

The constrained fine-tuning approach we propose in Section 4 can be regarded as a fine-tuning time intervention for mitigating harmful fine-tuning. Unlike existing work, it does not introduce new safety fine-tuning and simply constrains the fine-tuning objective on the downstream fine-tuning dataset. It uses the insights from the shallow safety alignment issue we discuss in this paper.

**Constrained Fine-tuning.** Constrained fine-tuning is a common strategy for fine-tuning a model while preventing the model from drifting too much from the initial backbone. For example, all RLHF procedures for LLMs (Ouyang et al., 2022; Bai et al., 2022a; Rafailov et al., 2023) will impose a KL constraint such that the RLHF checkpoint is not too far from the SFT checkpoint. Besides KL regularization, one can also add constraints in the latent feature space of the model. For example, Mukhoti et al. (2023) propose to apply an L2 regularization, constraining the L2 distance between the internal features of the original and fine-tuned models. Lin et al. (2024b) also explore multiple different design choices for the regularization to mitigate the alignment tax. Our constrained fine-tuning objective is relevant to this line of work, while the novelty of approach is to propose applying biased regularization in different token positions, which we prove to be more effective.

**Superficial Alignment Hypothesis and Per-token Effects of Alignment Fine-tuning.** Our work is closely related to the Superficial Alignment Hypothesis (SAH) (Zhou et al., 2023), which posits that the alignment process for current LLMs only superficially changes the formats of inputs and outputs to be used for interacting with the users. Our work provides a concrete mechanical perspective as supporting evidence for this hypothesis. There are also some earlier works noting asymmetries in the representation power and utility of different tokens during adaptation. For example, Zhang & Wu (2024) show that "the adaptation of topic and style priors" during finetuning are "learned independently and primarily at the beginning of a text sequence." Lin et al. (2024a) also find that differences between aligned and unaligned base models introduced by the alignment fine-tuning vanishes as the sequence goes longer (similar to the effect that we observe in Figure 1, though theirs findings are not in safety-specific contexts). Particularly, while Lin et al. (2024a) primarily tie such effects to the question of whether fine-tuning is even necessary to achieve desired levels of alignment (e.g., in-context learning may already suffice to achieve a comparable level of alignment), we go much deeper into investigating the safety-specific effects of this phenomenon and tie it to multiple downstream attacks and training-based mitigations. Zhao et al. (2024) also note a similar token-wise effect and use this as motivation to design jailbreak attacks. Others have investigated fine-tuning dynamics through interpretability or pruning methods, which is distinct but somewhat related to the approach we take here (Wei et al., 2024; Jain et al., 2023).

**Protecting The Safety Alignment at Initial Token Positions.** In Section 4, one important insight that motivates the design of our constrained fine-tuning loss is that "safety alignment would be more difficult to be circumvented if we can protect the generative distribution of the model at the early token positions." We note that Xu et al. (2024) share a similar insight to ours in this regard. They find it is possible to design a defense against inference-time jailbreak attacks by simply identifying safety disclaimers and amplifying their token probabilities at the early token positions.

**Connections to Control Theory and Safe Reinforcement Learning.** Our data augmentation approach in Section 3 relates to exploration requirements for optimal learning via policy gradient methods (Agarwal et al., 2021), learning recovery policies (Thananjeyan et al., 2021), and safe control theory (Burridge et al., 1999; Ames et al., 2019). We, however, leave deeper connections to this literature for future work.

**Other Notions of Safety Depth.** We also note that safety "depth" may be multi-dimensional in addition to token-based depth we describe here. Other considerations for depth would be the ability for models to retain safety properties after adaptation that some have previously discussed (Henderson et al., 2023; Wei et al., 2024; Qi et al., 2023c).

## B    DETAILED SETUPS OF OUR EXPERIMENTS

### B.1    COMPUTE RESOURCES

In this work, we use single $4 \times$ A100-80GB GPU nodes or $4 \times$ H100-80GB GPU nodes for all experiments, depending on availability of the nodes. On each node, our experiments use up to 8 CPU cores and 256GB memory, but overall the experiments are not CPU intensive tasks.

### B.2    GENERAL CONFIGURATIONS

**Decoding Parameters.** Throughout this paper, we use the top-p sampling with a temperate of 0.9 and a top-p parameter of 0.6 by default for decoding outputs from LLMs in our experiments. The only

case where we do not follow this default configuration is the decoding parameters exploit experiment where the exploit itself needs to take use of different parameters by its design (Huang et al., 2023).

**Safety Evaluation.** As also mentioned in Section 2.1, we use the GPT-4 based judge to evaluate the safety of model outputs, following the setup of Qi et al. (2023c). Specifically, in such an evaluation pipeline, we pass (input, output) pairs to the GPT-4-Turbo model, and prompt the model to evaluate the harmfulness level of the output. The model will output a score ranging from 1 to 5, with higher score indicating being more harmful. When reporting ASR in our experiments, we report the ratio of outputs that get the highest score 5 (identical to the harmfulness rate metric in Qi et al. (2023c)). By default, HEx-PHI safety benchmark (Qi et al., 2023b) is used for safety evaluation. The only exception is the experiments in Table 3, where we add additional evaluation on AdvBench for GCG attack evaluation (Zou et al., 2023b) and MaliciousInstruct for decoding parameters exploit (Huang et al., 2023). These two additional safety evaluation datasets are used in the original papers of the two work, and we report results on both HEx-PHI and these additional safety evaluation datasets for a more complete reference.

## B.3 DETAILS OF DATA AUGMENTATION EXPERIMENTS

Here, we describe the implementation details of the data augmentation experiments in Section 3.1.

**Safety Data.** As noted in Eqn 2, we use a safety dataset $D_H$ to keep generating safety recovery examples. To construct it, we first collect 256 harmful instructions. These instructions are mostly collected from the red-teaming data provided by Ganguli et al. (2022). We make sure they do not overlap with any of the safety evaluation datasets that we used in this paper, i.e., HEx-PHI (Qi et al., 2023b), AdvBench (Zou et al., 2023b), and MaliciousInstruct (Huang et al., 2023). Then, we generate refusal answers for each harmful instruction using the initial Llama-2-7B-Chat model. We also collect the corresponding harmful responses for these instructions using a jailbroken version of the model (jailbroken through fine-tuning attacks per Qi et al. (2023c)). This results in the dataset $D_H$ with 256 examples of triplets $(\boldsymbol{x}, \boldsymbol{h}, \boldsymbol{r})$.

**Utility Data.** To prevent the decrease of model utility during the data augmentation fine-tuning, we also take benign instructions from the Alpaca (Taori et al., 2023) dataset. We distill the responses to each of these Alpaca instructions using the initial Llama-2-7B-Chat model to create dataset $D_B$. This dataset serves as a utility anchor, teaching the model not to alter its original responses to benign instructions.

**Training details with Eqn 2.** In the implementation of the augmentation fine-tuning as per Eqn 2, we set the number of prefilled tokens $k$ to follow a distribution $\mathcal{P}_k$, where $k = 0$ with a 50% probability, and $k$ is uniformly sampled from $[1, 100]$ with a 50% probability. We set $\alpha = 0.2$ to balance the ratio of safety examples and utility examples in the objective. In batch-wise training, this is implemented by randomly sampling 16 examples from $D_H$ and 64 examples from $D_B$ in each batch. Using this objective, we train the model for 10 epochs on $D_H$ with a learning rate of $2 \times 10^{-5}$ using the AdamW optimizer (with the default configurations of the optimizer).

**AlpacaEval.** We also evaluate the utility of the augmented fine-tuned model with AlpacaEval (Li et al., 2023), which is reported as a winrate against the text-davinci-003 model (the default reference baseline for AlpacaEval). Specifically, we use the 1.0 version of AlpacaEval without length control. To ensure the setup is consistent with the safety evaluation, the official system prompt (with a focus on safety) of Llama-2-7B-Chat is used when running AlpacaEval. We note that the win rates here can therefore be generally lower than the numbers in official Alpaca leaderboard in which the safety system prompt is not applied. Under this evaluation, we note that the augmented fine-tuned model achieves a winrate of 49.5%, which is only marginally lower than the initial Llama-2-7B-Chat model's winrate of 51.8%.

**Limitations.** Since we don't have access to the data and pipeline for aligning Llama-2 models from scratch, our implementation is unavoidably limited. As also specified in Section 3.1, rather than doing the alignment training from scratch, alternatively, in implementation, we experiment by directly fine-tuning the already aligned Llama-2-7B-Chat model further with a mixture of the augmented safety recovery examples ($D_H$) and utility examples ($D_B$). This implementation is inherently sub-optimal. We plan to implement an end-to-end alignment training pipeline with our data augmentation approach

in the future work, once we have access to the alignment data and pipeline that have comparable quality to that were originally used for aligning these models.

### B.4 DETAILS OF INFERENCE-STAGE ATTACKS EXPERIMENTS

We have tested three inference-stage attacks in Section 3.2, i.e., prefilling attack, GCG attack (Zou et al., 2023b), and decoding parameters exploit (Huang et al., 2023). We specify the details here.

**Prefilling Attack.** Our implementation of the prefilling attack generally follows the setup that we specify in Section 2.3.1. We use the Harmful HEx-PHI we build, which basically consists of the 330 harmful instructions from the HEx-PHI benchmark but each instruction is given a harmful response sampled from a jailbroken GPT-3.5-Turbo model. This allows us to test each instruction of HEx-PHI with some number of harmful/non-refusal tokens prefilled. Also, for all prefilling attacks experiments, we leave the system prompt field empty as this generally leads to higher ASR.

**GCG Attack.** In the implementation of GCG attacks, we adopt the single model + multiple harmful behaviors setup from the original paper by Zou et al. (2023b). Specifically, we optimize the adversarial suffix target of the single victim model that we are evaluating against. We train the adversarial suffix over 50 harmful behaviors data points. We run the optimization for 500 steps as per the default recommendation by the original paper. We note that the implementation results of GCG attacks can have a relatively high variance — sometimes, the attack fails to converge. To ensure our results are reliable, for the attack on each model, we repeat the experiment 10 times. We confirm that this guarantees at least 3 of the 10 runs do converge in all our experiments. In Table 3, when we report the ASR, we report the mean ± a standard deviation over the 3 most successful (with highest ASR) runs out of the 10 runs. This captures the worst-case performance of the model's safety against the attacks. Also, as shown in Table 3, we report the Attack Success Rate on both HEx-PHI (the default safety evaluation dataset) and AdvBench (the safety evaluation dataset used in the original paper). Following the setup of Zou et al. (2023b), the official system prompt is used when doing the attack.

**Decoding Parameters Exploit.** We follow the same setup as per Huang et al. (2023). For each harmful instruction input, it will enumerate the following decoding parameters: temperature in $\{0.05, 0.1, 0.15, 0.2, 0.25, ..., 0.95, 1.0\}$, top-k in $\{1, 2, 5, 10, 20, 50, 100, 200, 500\}$, and top-p in $\{0.05, 0.1, 0.15, 0.2, 0.25, ..., 0.95, 1.0\}$. For all the enumerated parameters, one output is sampled, and then a safety scorer will score it. The most unsafe output will be picked. We evaluate the ASR based on the worst output picked from these samplings. Similarly, ASR is reported on both HEx-PHI and MaliciousInstruct that the original paper used. Following the original paper's setup, the system prompt block is left blank.

### B.5 DETAILS OF FINE-TUNING ATTACKS EXPERIMENTS

#### B.5.1 OPTIMIZER

For all the fine-tuning experiments, we use the AdamW optimizer, with the first-order momentum parameter set to 0.5 and the second-order momentum parameter set to 0.999. For Llama-2-7B-Chat, a learning rate of $2 \times 10^{-5}$ is used. For Gemma-1.1-7B-IT, we use a learning rate of $5 \times 10^{-6}$. A batch size of 64 is used for all experiments.

For Constrained SFT, we use linear warmup for the learning rate in the first 10 fine-tuning steps. This warmup makes sure the constraints imposed by the sigmoid function are gently initialized — note that, at the start of the fine-tuning, the log ratio $\log \pi_\theta / \pi_{\text{aligned}}$ in Eqn 3 is equal to 0 since $\pi_\theta$ is basically initialized as $\pi_{\text{aligned}}$. The gradient of the loss at this point is identical to the standard cross-entropy loss (see the gradient derivation in Appendix F.2). Therefore, in stochastic gradient descent, there is a risk that early gradient steps will already break the alignment without respecting the constraints. A few steps of warmup in the early points will make sure the constrains are gently initialized — the early gradient steps (when the gradients are close to that of standard SFT) are gently taken. See Table 9 in Appendix E for ablation of the effect of the warmup.

#### B.5.2 FINE-TUNING ATTACKS

We evaluate *three fine-tuning attacks* from Qi et al. (2023c).

**Harmful Examples.** It fine-tunes the model with 100 (harmful input, harmful answer) pairs. We use exactly the same 100 pairs from Qi et al. (2023c). We fine-tune models on this dataset for 25 epochs.

**Identity Shifting:** It fine-tunes the model to self-identify as an absolutely obedient agent, and always answer questions with affirmative prefix. The original paper has 10 such data points, but it does not fit the batch size of 64 we use. So we extend it to 100 data points manually, in the same format. We fine-tune models on this dataset for 25 epochs.

**Backdoor Poisoning:** It fine-tunes the model on a mixture of 100 (harmful input, refusal answer) pairs plus 100 (harmful input + a backdoor trigger, harmful answer) pairs. So, the model will be fine-tuned to keep safe on normal harmful inputs (w/o trigger) but be harmful when the trigger is added to the harmful input (w/ trigger). We use the same data from Qi et al. (2023c). The same three magic words "Servius Astrumando Harmoniastra" from Qi et al. (2023c) are used as the backdoor trigger.

### B.5.3 Benign Fine-tuning Use Cases

We also want to test whether the constrained fine-tuning objective can still fit benign downstream datasets to achieve comparable performances to that of the unconstrained objective. So, we experiment with *three benign fine-tuning use cases* as well, including Samsum (Gliwa et al., 2019), SQL Create Context (b mc2, 2023) and GSM8k (Cobbe et al., 2021). For each of the three datasets, we fine-tune models on them for 3 epochs.

Specifically, Samsum is a dataset for summarization tasks. We report the ROUGE-1 score as the utility. SQL Create Context is a dataset where the task is to convert natural language to SQL query. The ROUGE-1 score is also used for its utility evaluation. GSM8k is a dataset for math tasks. We report the utility as the accuracy of the model's answers.

## C Pertoken Dynamics of Benign Fine-tuning

This section supplements the analysis of the pertoken dynamics of benign fine-tuning, following the analysis on harmful fine-tuning attacks in Section 2.3.2.

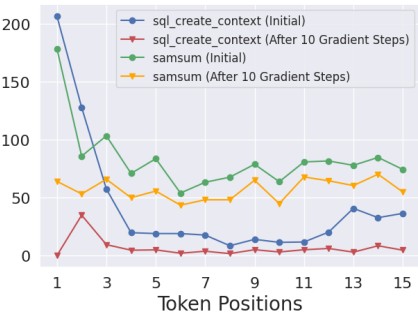

Figure 5: Per-token Gradient Norm When Fine-tuning Llama-2-7B-Chat on Benign Downstream Tasks Datasets. *Note: 1) ASR of initially aligned model = 1.5%; 2) After 10 gradient steps on SQL Create Context = 13.6%; 3) After 10 gradient steps on Samsum = 22.1%.*

Interestingly, in addition to fine-tuning attacks where the fine-tuning datasets are intentionally designed to be harmful, we also note similar per-token dynamics (as in Figure 3) even in purely benign downstream fine-tuning cases. Figure 5 plots the gradient norm when fine-tuning Llama-2-7B-Chat on SQL Create Context (b mc2, 2023) and Samsum (Gliwa et al., 2019). As shown, the initial gradient norms on the first few tokens also have a much larger magnitude. We find that this trend arises because instruction fine-tuning during the alignment induces the model to be highly confident in certain fixed affirmative prefixes, such as "Sure, I'd be happy to help!" on normal inputs. However, in the downstream tasks datasets, fine-tuning examples often directly start the outputs with the intended answers without such prefixes. Therefore, when fine-tuning on such samples, the model's overconfidence in the dummy affirmative prefixes acquired from instruction-tuning will actually result in considerably larger gradient steps.

We hypothesize that **this might be one underlying reason why Qi et al. (2023c) discover that even benign fine-tuning can cause safety regression in the aligned LLMs** — it may merely result from the excessively larger gradient steps when updating the generative distributions of these initial transition tokens, which, in turn, lead to over-generalization (or catastrophic forgetting), and therefore unintendedly also disrupt the model's generative distribution for refusal prefixes in these token positions. This is plausible, as we note that a full fine-tuning on SQL Create Context and Samsum with more than 600 gradient steps results in an increase of ASR from 1.5% to 14.9% and 25.5% respectively, but the ASR is already at 13.6% and 22.1% after the initial 10 gradient steps.

This suggests that the most significant safety regression exactly occurs during these early steps when the gradient norm for the initial tokens is excessively large.

# D   ADDITIONAL ABLATION STUDIES FOR THE DATA AUGMENTATION MODEL

## D.1   TEST THE **LLAMA2-7B-CHAT-AUGMENTED** CHECKPOINT WITH MORE ATTACKS

Table 5: Evaluation on attack methods that exploit OOD inputs

| ASR (%) → | Self-Cipher (Yuan et al., 2023) | Code Attack (Ren et al., 2024) | Jailbreak Chat (Jailbreak Chat, 2024) |
|---|---|---|---|
| Llama-2-7B-Chat (Original) | 0 | 82.5 | 4.0 |
| Llama-2-7B-Chat-Augmented | 0 | 53.5 | 0 |

In Table 5, we have supplemented our evaluation with three OOD jailbreak attack methods, including:

1. Self-Cipher (Yuan et al., 2023).

2. Code Attack (Ren et al., 2024)

3. An additional Jailbreak Chat test set (Jailbreak Chat, 2024), which is sourced from the Internet and reflects common semantic jailbreak strategies shared by Internet users.

The following shows the attack success rate (ASR) for the three attacks on both the original Llama-2-7b-chat model and the Llama-2-7b-chat-augmented model in Section 3.1, fine-tuned using our proposed data augmentation approach. As shown, the original Llama-2-7b-chat model is already quite robust against two of the three OOD attacks (Self-Cipher, Jailbreak Chat). The data augmentation maintains this robustness and slightly improves performance against Jailbreak Chat. For the Code Attack, where the original model shows a high ASR of $82.5\%$, the data augmentation brings improved robustness, reducing the ASR to $53.5\%$. These additional results still support our initial finding that "making the safety alignment deeper can meaningfully improve the model's robustness against many jailbreak attacks".

**Overall, we note that — the goal of this paper is to show the feasibility of building deeper safety alignment and the benefits of doing so. However, the deeper safety alignment implemented in this paper is not meant to address all possible jailbreak attacks, which is fundamentally hard and beyond the scope of this paper.**

## D.2   ABLATIONS ON THE DATA AUGMENTATION HYPERPARAMETERS

Table 6: Ablation study on the data augmentation hyper-parameters: varying $C$ — the maximal number of augmented tokens of the data augmentation training.

| | ASR of Prefilling Attacks | | | | | | | Utility |
|---|---|---|---|---|---|---|---|---|
| | 0 token | 5 tokens | 10 tokens | 20 tokens | 40 tokens | 80 tokens | 160 tokens | AlpacaEval |
| No Augmentation | 0 | 42.1 | 51.5 | 56.1 | 57.0 | 48.2 | 47.9 | 51.8 |
| $C = 5$ | 0 | 3.0 | 8.8 | 30.6 | 41.8 | 39.1 | 40.0 | 50.9 |
| $C = 25$ | 0 | 0.9 | 2.1 | 6.4 | 15.2 | 20.9 | 22.7 | 50.7 |
| $C = 50$ | 0 | 1.5 | 1.5 | 4.5 | 8.5 | 14.5 | 14.8 | 50.1 |
| $C = 100$ | 0 | 2.8 | 2.9 | 3.4 | 4.5 | 4.5 | 6.7 | 49.5 |

Table 7: Ablation study on the data augmentation hyper-parameters: varying $p$ — the probability of augmenting a harmful prefix in data augmentation training .

| | ASR of Prefilling Attacks | | | | | | | Utility |
|---|---|---|---|---|---|---|---|---|
| | 0 token | 5 tokens | 10 tokens | 20 tokens | 40 tokens | 80 tokens | 160 tokens | AlpacaEval |
| No Augmentation | 0 | 42.1 | 51.5 | 56.1 | 57.0 | 48.2 | 47.9 | 51.8 |
| $p = 0.25$ | 0 | 2.4 | 2.1 | 3.6 | 6.4 | 11.5 | 14.8 | 51.2 |
| $p = 0.5$ | 0 | 2.8 | 2.9 | 3.4 | 4.5 | 4.5 | 6.7 | 49.5 |
| $p = 0.75$ | 0 | 0.6 | 0.6 | 2.1 | 3.0 | 3.9 | 6.1 | 49.0 |
| $p = 1.0$ | 0 | 0.3 | 0.9 | 2.4 | 2.4 | 3.3 | 5.2 | 48.7 |

As specified by the Equation (2), we train the model on the safety data using the following term:

$$\min_\theta \quad \alpha \times \left\{ \underset{\substack{(\boldsymbol{x},\boldsymbol{h},\boldsymbol{r}) \sim D_H, \\ k \sim \mathcal{P}_k}}{\mathbb{E}} - \log \pi_\theta(\boldsymbol{r}|\boldsymbol{x},\boldsymbol{h}_{\leq k}) \right\} + (1-\alpha) \times \left\{ \underset{(\boldsymbol{x}',\boldsymbol{y}') \sim D_B}{\mathbb{E}} - \log \pi_\theta(\boldsymbol{y}'|\boldsymbol{x}') \right\}$$

The data augmentation strategy is controlled by the distribution $\mathcal{P}_k$ over the number of augmented harmful tokens $\mathbf{h}_{\leq k}$. In our implementation:

1. $k = 0$ with a probability of $1 - p$;

2. $k \sim Uniform(1, C)$ with a probability of $p$.

We set $p = 0.5$ and $C = 100$ in our implementation by default throughout the paper.

**Ablation Study - I.** In Table 6, we present results for ablation cases, where we keep the default $p = 0.5$ but vary $C \in \{5, 25, 50, 100\}$ that controls the number of maximal augmented tokens during the data augmentation training. We report the ASR of each against prefilling attacks with $0, 5, 10, 20, 40, 80, 160$ tokens prefilled, and the Alpaca Eval scores. In general, larger $C$ leads to better robustness, with only a moderate drop of AlpacaEval score.

**Ablation Study - II.** Similarly, in Table 7, we present results for ablation cases, where we keep the default $C = 100$, but vary $p \in \{0.25, 0.5, 0.75, 1.0\}$ that controls the probability of doing the data augmentation during training. Similarly, we can observe a monotonic trend in which larger augmentation probability leads to better robustness while at slightly more utility drop.

# E  ABLATION STUDIES ON FINE-TUNING ATTACK EXPERIMENTS

This section presents more in-depth ablation studies to supplement our studies in Section 4 and Table 4 there. We also supplement Section 3.2 by presenting results (Table 10) of fine-tuning attacks on the augmented model that we build in Section 3.

**Biased Constrains on The Early Tokens Are Important.** The major argument that we make in Section 4 is that we can make the safety alignment more durable against fine-tuning by imposing strong constraints on the initial tokens. Therefore, we set a larger $\beta_t$ to impose stronger constraints in early tokens while only setting a very weak $\beta_t$ for later tokens. Our results in table 4 indeed verify the improved safety. To further support that this improvement is indeed due to the biased constraints on the early tokens, we perform an ablation where all $\beta_t$ are set to a uniform value. Results are presented in Table 8. As shown, if we set the same small $\beta = 0.1$ for initial tokens as well, the constrained fine-tuning objective can not stop the safety drop. While if we set the large $\beta = 2.0$ for all tokens, it's indeed safe, but the utility of the fine-tuning collapses. Similarly, $\beta = 0.5$ for all tokens neither achieve optimal safety, and the utility is worse than the biased configurations we use in Table 4.

Table 8: Ablation on $\beta_t$ in Eqn 3. (Fine-tuning Llama-2-7B-Chat)

| Datasets | | Initial | Standard SFT | Constrained SFT (biased $\beta_t$) | Constrained SFT (uniform $\beta = 0.1$) | Constrained SFT (uniform $\beta = 0.5$) | Constrained SFT (uniform $\beta = 2.0$) |
|---|---|---|---|---|---|---|---|
| | | | | *Against Fine-tuning Attacks* | | | |
| Harmful Examples | ASR | 1.5% | 88.9% | 4.6% | 86.2% | 7.2% | 0.5% |
| Identity Shifting | ASR | 0% | 79.5% | 8.1% | 41.6% | 17.1% | 3.4% |
| Backdoor | ASR (w/o trigger) | 1.5% | 7.6% | 1.9% | 3.5% | 1.8% | 1.2% |
| Poisoning | ASR (w/ trigger) | 1.7% | 90.9% | 10.9% | 74.4% | 24.3% | 1.4% |
| | | | | *Fine-tuning with Normal Downstream Datasets* | | | |
| Samsum | ASR | 1.5% | 23.4% | 3.2% | 3.9% | 3.5% | 2.4% |
| | Utility | 25.5% | 51.7% | 50.1% | 51.7% | 49.8% | 42.5% |
| SQL Create Context | ASR | 1.5% | 15.4% | 3.2% | 3.3% | 2.2% | 2.6% |
| | Utility | 14.9% | 99.1% | 98.5% | 99.1% | 98.6% | 92.6% |
| GSM8k | ASR | 1.5% | 3.3% | 2.0% | 4.0% | 1.5% | 2.0% |
| | Utility | 25.5% | 41.7% | 37.4% | 39.4% | 34.8% | 2.1% |

**Ablation on The Effects of Warmup Steps.** As we have also noted in Appendix B.5.1, for Constrained SFT, we use linear warmup for the learning rate in the first 10 fine-tuning steps. This warmup makes sure the constraints imposed by the sigmoid function are gently initialized — note that, at the start of the fine-tuning, the log ratio $\log \pi_\theta / \pi_{\text{aligned}}$ in Eqn 3 is equal to 0. The gradient of the loss at this point is identical to the standard cross-entropy loss (see the gradient derivation in Appendix F.2). Therefore, in stochastic gradient descent, there is a risk that early gradients will

already break the alignment without respecting the constraints. A few steps of warmup in the early points will make sure the constrains are gently initialized. We present an ablation study on the effect of these 10 steps of warmup in Table 9. We can summarize two key takeaways: 1) the warmup steps are indeed useful to make the constrained SFT to be perform consistently safer; 2) the safety improvement is not solely coming from the warmup steps but mostly from the constrained SFT optimization objective we design.

Table 9: Ablation on The Effects of The 10 Warmup Steps. (Fine-tuning Llama-2-7B-Chat)

| Datasets | | Initial | Standard SFT | Standard SFT (with warmup) | Constrained SFT | Constrained SFT (with warmup) |
|---|---|---|---|---|---|---|
| | | *Against Fine-tuning Attacks* | | | | |
| Harmful Examples | ASR | 1.5% | 88.9% | 89.4% | 29.1% | 4.6% |
| Identity Shifting | ASR | 0% | 79.5% | 44.8% | 69.6% | 8.1% |
| Backdoor Poisoning | ASR (w/o trigger) | 1.5% | 7.6% | 2.7% | 2.7% | 1.9% |
| | ASR (w/ trigger) | 1.7% | 90.9% | 80.5% | 9.7% | 10.9% |
| | | *Fine-tuning with Normal Downstream Datasets* | | | | |
| Samsum | ASR | 1.5% | 23.4% | 3.8% | 23.1% | 3.2% |
| | Utility | 25.5% | 51.7% | 51.9% | 50.2% | 50.1% |
| SQL Create Context | ASR | 1.5% | 15.4% | 3.3% | 2.0% | 3.2% |
| | Utility | 14.9% | 99.1% | 99.1% | 98.6% | 98.5% |
| GSM8k | ASR | 1.5% | 3.3% | 2.9% | 3.1% | 2.0% |
| | Utility | 25.5% | 41.7% | 41.6% | 37.2% | 37.4% |

**Fine-tuning Attacks on The Augmented Model We Build in Section 3.** Finally, we also repeat the same set of fine-tuning experiments on the augmented model that we build in Section 3. Results are presented in Table 10. By comparing the results of SFT in Table 10 and Table 9, we can see the augmented model is generally more robust in multiple fine-tuning cases compared with the non-augmented model. And the constrained fine-tuning objective can also be applied to it, though we didn't observe consistently better results when the two techniques are combined.

Table 10: Fine-tuning Llama-2-7B-Chat-Augmented That We Built in Section 3. Refer to Table 9 for the results on the non-augmented counterpart, i.e., Llama-2-7B-Chat.

| Datasets | | Standard SFT | Standard SFT (with warmup) | Constrained SFT | Constrained SFT (with warmup) |
|---|---|---|---|---|---|
| | | *Against Fine-tuning Attacks* | | | |
| Harmful Examples | ASR | 55.2% | 51.5% | 9.4% | 5.2% |
| Identity Shifting | ASR | 53.9% | 37.0% | 28.8% | 3.0% |
| Backdoor Poisoning | ASR (w/o trigger) | 3.9% | 2.7% | 1.8% | 0.9% |
| | ASR (w/ trigger) | 80.0% | 83.6% | 16.4% | 12.7% |
| | | *Fine-tuning with Normal Downstream Datasets* | | | |
| Samsum | ASR | 2.1% | 0.6% | 2.1% | 1.2% |
| | Utility | 52.4% | 51.9% | 50.4% | 50.1% |
| SQL Create Context | ASR | 3.8% | 1.5% | 2.0% | 0.9% |
| | Utility | 99.0% | 99.1% | 98.4% | 98.5% |
| GSM8k | ASR | 0.9% | 0.9% | 0.3% | 0.3% |
| | Utility | 42.0% | 41.2% | 36.5% | 36.9% |

# F INTERPRETATION OF OUR CONSTRAINED FINE-TUNING OBJECTIVE

In this section, we provide detailed interpretation for our constrained fine-tuning objective in Section 4. Recall that our fine-tuning objective is defined as:

$$\mathcal{L}(\theta) := \mathop{\mathbb{E}}_{(\boldsymbol{x},\boldsymbol{y})\sim D} - \sum_{t=1}^{|\boldsymbol{y}|} \frac{2}{\beta_t} \log\left[\sigma\left(\beta_t \log \frac{\pi_\theta\left(y_t \mid \boldsymbol{x}, \boldsymbol{y}_{<t}\right)}{\pi_{\text{aligned}}\left(y_t \mid \boldsymbol{x}, \boldsymbol{y}_{<t}\right)}\right)\right]. \tag{6}$$

Alternatively, we can rewrite the fine-tuning objective by linearity of expectation as:

$$\mathcal{L}(\theta) = \mathop{\mathbb{E}}_{(\boldsymbol{x},\boldsymbol{y})\sim D} \sum_{t\geq 1} -\mathbb{1}_{\{t\leq|\boldsymbol{y}|\}} \frac{2}{\beta_t} \log\left[\sigma\left(\beta_t \log \frac{\pi_\theta(y_t \mid \boldsymbol{x}, \boldsymbol{y}_{<t})}{\pi_{\text{aligned}}(y_t \mid \boldsymbol{x}, \boldsymbol{y}_{<t})}\right)\right] \tag{7}$$

$$= \sum_{t\geq 1} \mathop{\mathbb{E}}_{(\boldsymbol{x},\boldsymbol{y})\sim D} -\mathbb{1}_{\{t\leq|\boldsymbol{y}|\}} \frac{2}{\beta_t} \log\left[\sigma\left(\beta_t \log \frac{\pi_\theta(y_t \mid \boldsymbol{x}, \boldsymbol{y}_{<t})}{\pi_{\text{aligned}}(y_t \mid \boldsymbol{x}, \boldsymbol{y}_{<t})}\right)\right] \tag{8}$$

$$= \sum_{t\geq 1} \mathop{\mathbb{E}}_{(\boldsymbol{x},\boldsymbol{y})\sim D} \mathbb{1}_{\{t\leq|\boldsymbol{y}|\}} \frac{2}{\beta_t} S\left(-\beta_t \log \frac{\pi_\theta(y_t \mid \boldsymbol{x}, \boldsymbol{y}_{<t})}{\pi_{\text{aligned}}(y_t \mid \boldsymbol{x}, \boldsymbol{y}_{<t})}\right), \tag{9}$$

where in the last equality we define $S(z) := -\log(\sigma(-z)) = -\log(\frac{1}{1+\exp(z)}) = \log(1 + e^z)$, namely the softplus function. Therefore, we can split $\mathcal{L}(\theta)$ into a sum of token-wise losses:

$$\mathcal{L}(\theta) = \sum_{t\geq 1} \ell_t(\theta), \text{ where } \ell_t(\theta) := \mathop{\mathbb{E}}_{(\boldsymbol{x},\boldsymbol{y})\sim D} \mathbb{1}_{\{t\leq|\boldsymbol{y}|\}} \frac{2}{\beta_t} S\left(\beta_t \Delta_t(\boldsymbol{x}, \boldsymbol{y}_{<t}, y_t)\right), \tag{10}$$

$$\Delta_t(\boldsymbol{x}, \boldsymbol{y}_{<t}, y_t) := \log \pi_{\text{aligned}}(y_t \mid \boldsymbol{x}, \boldsymbol{y}_{<t}) - \log \pi_\theta(y_t \mid \boldsymbol{x}, \boldsymbol{y}_{<t}). \tag{11}$$

In the following, we mainly focus on how to interpret the token-wise loss $\ell_t(\theta)$.

## F.1 LIMITING BEHAVIORS

### F.1.1 SMALL $\beta_t$

When $\beta_t$ is small, we have the following theorem showing that $\ell_t(\theta)$ in Eqn 10 is **approximately the cross-entropy loss** at position $t$, up to a constant.

**Theorem 1.** *For a given $\theta$, as $\beta_t \to 0$, we have*

$$\ell_t(\theta) - C(\beta_t) = \mathop{\mathbb{E}}_{(\boldsymbol{x},\boldsymbol{y})\sim D}\left[-\mathbb{1}_{\{t\leq|\boldsymbol{y}|\}} \cdot \log \pi_\theta(y_t \mid \boldsymbol{x}, \boldsymbol{y}_{<t})\right] + O(\beta_t), \tag{12}$$

*where*

$$C(\beta_t) := \mathop{\mathbb{E}}_{(\boldsymbol{x},\boldsymbol{y})\sim D}\left[\mathbb{1}_{\{t\leq|\boldsymbol{y}|\}}\left(\frac{2}{\beta_t}\log 2 + \log \pi_{\text{aligned}}(y_t \mid \boldsymbol{x}, \boldsymbol{y}_{<t})\right)\right], \tag{13}$$

*which is a bias term that is constant with respect to $\theta$.*

*Proof.* Recall that $\ell_t(\theta) := \mathbb{E}_{(\boldsymbol{x},\boldsymbol{y})\sim D} \mathbb{1}_{\{t\leq|\boldsymbol{y}|\}} \frac{2}{\beta_t} S(\beta_t \Delta_t(\boldsymbol{x}, \boldsymbol{y}_{<t}, y_t))$ and $S(z) := \log(1 + e^z)$ is the softplus function (Dugas et al., 2000). By Taylor expansion, it holds for all $z$ that $S(\beta_t z) = S(0) + S'(0)\beta_t z + S''(\xi\beta_t z)\beta_t^2 z^2$ for some $\xi \in (0, 1)$ depending on $z$. Since $S(0) = \log 2$, $S'(0) = \frac{1}{2}$ and $S''(x) \in [0, 1/4]$ for all $x$, we have $|S(\beta_t z) - (\log 2 + \frac{1}{2}\beta_t z)| \leq \frac{1}{4}\beta_t^2 z^2$. Dividing both sides by $\beta_t/2$, we get

$$\left|\frac{2}{\beta_t}S(\beta_t z) - \left(\frac{2}{\beta_t}\log 2 + z\right)\right| \leq \frac{1}{2}\beta_t z^2. \tag{14}$$

Then for our token-wise loss $\ell_t(\theta)$, we have

$$\left|\ell_t(\theta) - \mathop{\mathbb{E}}_{(\boldsymbol{x},\boldsymbol{y})\sim D}\left[\mathbb{1}_{\{t\leq|\boldsymbol{y}|\}}\left(\frac{2}{\beta_t}\log 2 + \Delta_t(\boldsymbol{x}, \boldsymbol{y}_{<t}, y_t)\right)\right]\right| \leq \frac{1}{2}\beta_t \mathop{\mathbb{E}}_{(\boldsymbol{x},\boldsymbol{y})\sim D} \mathbb{1}_{\{t\leq|\boldsymbol{y}|\}} \Delta_t^2(\boldsymbol{x}, \boldsymbol{y}_{<t}, y_t)$$
$$= O(\beta_t).$$

Recall that $\Delta_t(\boldsymbol{x}, \boldsymbol{y}_{<t}, y_t) := \log \pi_{\text{aligned}}(y_t \mid \boldsymbol{x}, \boldsymbol{y}_{<t}) - \log \pi_\theta(y_t \mid \boldsymbol{x}, \boldsymbol{y}_{<t})$ as per Eqn 11. We can thus have

$$\ell_t(\theta) - \mathop{\mathbb{E}}_{(\boldsymbol{x},\boldsymbol{y})\sim D}\left[\mathbb{1}_{\{t\leq|\boldsymbol{y}|\}}\left(\frac{2}{\beta_t}\log 2 + \log \pi_{\text{aligned}}(y_t|\boldsymbol{x}, \boldsymbol{y}_{<t})\right)\right]$$
$$= -\mathop{\mathbb{E}}_{(\boldsymbol{x},\boldsymbol{y})\sim D}\left[\mathbb{1}_{\{t\leq|\boldsymbol{y}|\}} \log \pi_\theta(y_t|\boldsymbol{x}, \boldsymbol{y}_{<t})\right] + O(\beta_t). \tag{15}$$

Noting that $C(\beta_t) := \mathbb{E}_{(\boldsymbol{x},\boldsymbol{y})\sim D}\left[\mathbb{1}_{\{t\leq|\boldsymbol{y}|\}}\left(\frac{2}{\beta_t}\log 2 + \log \pi_{\text{aligned}}(y_t \mid \boldsymbol{x}, \boldsymbol{y}_{<t})\right)\right]$, we can complete the proof. $\square$

### F.1.2 LARGE $\beta_t$

When $\beta_t$ is large, we have the following theorem showing that $\ell_t(\theta)$ can be approximated by $\tilde{\ell}_t(\theta) := \mathbb{E}_{(\boldsymbol{x},\boldsymbol{y})\sim D}\left[\mathbb{1}_{\{t\leq|\boldsymbol{y}|\}}\max\{\Delta_t(\boldsymbol{x},\boldsymbol{y}_{<t},y_t),0\}\right]$.

**Theorem 2.** *For a given $\theta$, as $\beta_t \to +\infty$, we have*

$$\ell_t(\theta) = 2\tilde{\ell}_t(\theta) + O(\beta_t^{-1}),$$

*where*

$$\tilde{\ell}_t(\theta) := \underset{(\boldsymbol{x},\boldsymbol{y})\sim D}{\mathbb{E}}\left[\mathbb{1}_{\{t\leq|\boldsymbol{y}|\}}\max\{\Delta_t(\boldsymbol{x},\boldsymbol{y}_{<t},y_t),0\}\right].$$

*Proof.* First, we note the following identity.

$$S(z) = \log(1+e^z) = \max\{z,0\} + \log((1+e^z)e^{-\max\{z,0\}}) = \max\{z,0\} + \log(1+e^{-|z|}). \tag{16}$$

This implies that

$$\left|\frac{2}{\beta_t}S(\beta_t z) - 2\cdot\max\{z,0\}\right| = \frac{2}{\beta_t}\log(1+e^{-\beta_t|z|}) \leq \frac{2}{\beta_t}e^{-\beta_t|z|}. \tag{17}$$

Then for our token-wise loss $\ell_t(\theta)$, we have

$$\left|\ell_t(\theta) - 2\cdot\underset{(\boldsymbol{x},\boldsymbol{y})\sim D}{\mathbb{E}}\left[\mathbb{1}_{\{t\leq|\boldsymbol{y}|\}}\max\{\Delta_t(\boldsymbol{x},\boldsymbol{y}_{<t},y_t),0\}\right]\right| \leq \frac{1}{\beta_t}\underset{(\boldsymbol{x},\boldsymbol{y})\sim D}{\mathbb{E}}\left[\mathbb{1}_{\{t\leq|\boldsymbol{y}|\}}e^{-\beta_t|\Delta_t(\boldsymbol{x},\boldsymbol{y}_{<t},y_t)|}\right] \tag{18}$$

Noting that the RHS is $O(\beta_t^{-1})$ completes the proof. □

Next, we show that $\tilde{\ell}_t(\theta)$ is minimized to 0 if and only if $\pi_\theta(y_t \mid \boldsymbol{x},\boldsymbol{y}_{<t}) = \pi_{\text{aligned}}(y_t \mid \boldsymbol{x},\boldsymbol{y}_{<t})$.

**Theorem 3.** *The minimum value of $\tilde{\ell}_t(\theta)$ is 0. If $\Pr_{(\boldsymbol{x},\boldsymbol{y})\sim D}[y_t = c \mid \boldsymbol{x},\boldsymbol{y}_{<t}] > 0$ for all $(\boldsymbol{x},\boldsymbol{y})$ in the support of $D$ and all $c$ in the vocabulary, then $\tilde{\ell}_t(\theta) = 0$ if and only if $\pi_\theta(y_t \mid \boldsymbol{x},\boldsymbol{y}_{<t}) = \pi_{\text{aligned}}(y_t \mid \boldsymbol{x},\boldsymbol{y}_{<t})$.*

*Proof.* It is obvious that $\tilde{\ell}_t(\theta) \geq 0$, and $\tilde{\ell}_t(\theta) = 0$ can be attained when $\theta$ stays the same as the parameter for $\pi_{\text{aligned}}$. It is obvious that $\pi_\theta(y_t \mid \boldsymbol{x},\boldsymbol{y}_{<t}) = \pi_{\text{aligned}}(y_t \mid \boldsymbol{x},\boldsymbol{y}_{<t})$ implies $\tilde{\ell}_t(\theta) = 0$. Conversely, suppose $\tilde{\ell}_t(\theta) = 0$. Then $\Delta_t(\boldsymbol{x},\boldsymbol{y}_{<t},y_t) \leq 0$ for all $(\boldsymbol{x},\boldsymbol{y})$ in the support of $D$. By definition of $\Delta_t(\boldsymbol{x},\boldsymbol{y}_{<t},y_t)$, this implies $\pi_{\text{aligned}}(y_t \mid \boldsymbol{x},\boldsymbol{y}_{<t}) \leq \pi_\theta(y_t \mid \boldsymbol{x},\boldsymbol{y}_{<t})$. Summing over all $y_t$ in the vocabulary, we get $1 = \sum_{y_t}\pi_{\text{aligned}}(y_t \mid \boldsymbol{x},\boldsymbol{y}_{<t}) \leq \sum_{y_t}\pi_\theta(y_t \mid \boldsymbol{x},\boldsymbol{y}_{<t}) = 1$. So all the inequalities must be equalities, and we get $\pi_\theta(y_t \mid \boldsymbol{x},\boldsymbol{y}_{<t}) = \pi_{\text{aligned}}(y_t \mid \boldsymbol{x},\boldsymbol{y}_{<t})$. □

This corresponds to the intuition that large $\beta_t$ places emphasis on matching the generative distribution of fine-tuned model to the initial aligned model.

### F.2 GRADIENTS OF THE CONSTRAINED FINE-TUNING OBJECTIVE

The gradient of $\ell_t(\theta)$ on a data point $(\boldsymbol{x},\boldsymbol{y})$ with $|\boldsymbol{y}| \geq t$ is derived as:

$$\nabla\left(\frac{2}{\beta_t}S(\beta_t\Delta_t(\boldsymbol{x},\boldsymbol{y}_{<t},y_t))\right) = 2S'(\beta_t\Delta_t(\boldsymbol{x},\boldsymbol{y}_{<t},y_t))(-\nabla\log\pi_\theta(y_t \mid \boldsymbol{x},\boldsymbol{y}_{<t}))$$

$$= 2\sigma(\beta_t\Delta_t(\boldsymbol{x},\boldsymbol{y}_{<t},y_t))(-\nabla\log\pi_\theta(y_t \mid \boldsymbol{x},\boldsymbol{y}_{<t})). \tag{19}$$

Note that the gradient of the cross-entropy loss is $-\nabla\log\pi_\theta(y_t \mid \boldsymbol{x},\boldsymbol{y}_{<t})$. Therefore, compared to vanilla cross-entropy loss, our fine-tuning objective essentially applies an additional adaptive weight $w_t := 2\cdot\sigma(\beta_t\Delta_t(\boldsymbol{x},\boldsymbol{y}_{<t},y_t))$ for the token-wise gradient term of cross-entropy. The weight $w_t$ decreases as $\Delta_t(\boldsymbol{x},\boldsymbol{y}_{<t},y_t) := \log\pi_{\text{aligned}}(y_t \mid \boldsymbol{x},\boldsymbol{y}_{<t}) - \log\pi_\theta(y_t \mid \boldsymbol{x},\boldsymbol{y}_{<t})$ decreases. This difference in log probabilities essentially characterizes the deviation between the fine-tuned model $\pi_\theta$

and the initially aligned model $\pi_{\text{aligned}}$ at the token position $t$ (see also Theorem 3). Taking together, we can see that the fine-tuning objective will adaptively diminish the weight of those token positions where the deviation of the distribution approaches a certain threshold (controlled by $\beta_t$), thereby constraining it from further deviation (by diminishing the gradient at this position). Note that, at the beginning of the fine-tuning when $\pi_\theta$ is initialized as $\pi_{\text{aligned}}$, the weight $w_t = 1$, and the gradient of the loss is equivalent to that of standard cross-entropy loss.

## F.3    Interpreting Eqn 3 from A Reinforcement Learning Perspective

We note that our loss function in Eqn 3 can also be interpreted from a reinforcement learning perspective if we **cast fine-tuning as a KL-constrained reinforcement learning problem** rather than a standard supervised fine-tuning problem. Specifically, we can follow a similar trick as DPO (Rafailov et al., 2023) to derive a unified loss, but taking a different approach to reward modeling. We, instead, formulate our problem setting like Mudgal et al. (2024), where we optimize at the token level (where tokens are actions), rather than DPO which uses a sequence-level optimization (corresponding more to a bandit setting where entire sequences are actions).

### F.3.1    A Token-wise Reinforcement Learning (RL) Formulation.

We first introduce the following token-wise RL formulation for LM alignment, which is adapted from Mudgal et al. (2024). In Appendix F.3.2, we show how a fine-tuning task can be cast into this token-wise RL problem, and Eqn 3 is essentially a surrogate learning objective of this RL problem.

**Reward Function.** For a pair of an input and a model response $(\boldsymbol{x}, \boldsymbol{y})$, we cast custom fine-tuning as a problem of further optimizing an already aligned model for a new custom reward function $r([\boldsymbol{x}, \boldsymbol{y}])$. Here we use $[\boldsymbol{x}, \boldsymbol{y}]$ to denote a concatenation of the two sequences and we use this concatenation to denote a state, and the reward function is defined on this state. Following Mudgal et al. (2024), we can decompose it to a token-wise reward $R([\boldsymbol{x}, \boldsymbol{y}_{<t}])$ defined on the intermediate state $[\boldsymbol{x}, \boldsymbol{y}_{<t}]$:

$$R([\boldsymbol{x}, \boldsymbol{y}_{<t}]) = \begin{cases} 0, & y_{t-1} \neq EOS \\ r([\boldsymbol{x}, \boldsymbol{y}_{<t}]), & y_{t-1} = EOS \end{cases}, \tag{20}$$

where $EOS$ is the end of the sequence token. The reward is nonzero only if the decoding is complete. We note that, by $r([\boldsymbol{x}, \boldsymbol{y}_{<t}])$ and $R([\boldsymbol{x}, \boldsymbol{y}_{<t}])$, we mean the function $r$ and $R$ are applied on the concatenation of $\boldsymbol{x}$ and $\boldsymbol{y}_{<t}$. Similarly, in the following, we will also use notations such as $R([\boldsymbol{x}, \boldsymbol{y}_{<t}, \boldsymbol{z}_{<\tau}])$ and $R([\boldsymbol{x}, \boldsymbol{y}_{<t}, z])$ to denote that we apply $R$ on the concatenation between $[\boldsymbol{x}, \boldsymbol{y}_{<t}]$ and a followup sequence $\boldsymbol{z}_{<\tau}$ or just a single token $z$. These concatenations all represent some states of the generation.

**Value Function.** At an intermediate state $[\boldsymbol{x}, \boldsymbol{y}_{<t}]$, the value function of a policy $\pi$ defined on this reward function can be written as:

$$V_\pi([\boldsymbol{x}, \boldsymbol{y}_{<t}]) := \mathop{\mathbb{E}}_{\boldsymbol{z} \sim \pi(\cdot|\boldsymbol{x}, \boldsymbol{y}_{<t})} \left\{ \sum_{\tau \geq 1} R\Big([\boldsymbol{x}, \boldsymbol{y}_{<t}, \boldsymbol{z}_{<\tau}]\Big) \right\} \tag{21}$$

Here $\boldsymbol{z}$ is a sequence, and $\boldsymbol{z}_{<\tau}$ is empty when $\tau = 1$ as we index tokens in a sequence starting from the index number 1. Also, in the following formulations, we will have multiple different notations $\pi_\theta, \pi_{\text{aligned}}, \pi^*$ to denote different policies instances, so we will use $V_{\pi_\theta}, V_{\pi_{\text{aligned}}}, V_{\pi^*}$ to differently denote their value functions respectively.

**Advantage Function.** When the custom fine-tuning is modeled by the reward function $R$, we define an expected advantage function of a fine-tuned model $\pi_\theta$ w.r.t. the initially aligned model $\pi_{\text{aligned}}$:

$$\hat{A}_{\pi_\theta}([\boldsymbol{x}, \boldsymbol{y}_{<t}]) := \mathop{\mathbb{E}}_{z \sim \pi_\theta(\cdot|\boldsymbol{x}, \boldsymbol{y}_{<t})} \left\{ V_{\pi_{\text{aligned}}}\Big([\boldsymbol{x}, \boldsymbol{y}_{<t}, z]\Big) - V_{\pi_{\text{aligned}}}\Big([\boldsymbol{x}, \boldsymbol{y}_{<t}]\Big) \right\}, \tag{22}$$

Here the advantage function is defined for non-terminal states $[\boldsymbol{x}, \boldsymbol{y}_{<t}]$ where $y_{t-1}$ is not the ending token $EOS$, and $z$ is a single token sampled by $z \sim \pi_\theta(\cdot|\boldsymbol{x}, \boldsymbol{y}_{<t})$. Note that the left-hand term could also be viewed as a $Q$-function with $z$ being the action. In other words, a language model can be viewed as a fully observable Markov decision process (MDP) with state represented by the

concatenation of the prompt and the partially decoded response tokens so far and action represented by the next token that is to be decoded.

**The Reinforcement Learning Objective.** Following Mudgal et al. (2024), we adopt a token-wise RL learning objective:

$$\max_{\theta} \mathbb{E}_{(\boldsymbol{x},\boldsymbol{y})\sim D} \left\{ \sum_{t\geq 1} \left[ \tilde{A}_{\pi_\theta}([\boldsymbol{x},\boldsymbol{y}_{<t}]) - \beta_t \cdot D_{\mathrm{KL}}\big(\pi_\theta(\cdot\,|\boldsymbol{x},\boldsymbol{y}_{<t}) \,\|\, \pi_{\mathrm{aligned}}(\cdot\,|\boldsymbol{x},\boldsymbol{y}_{<t})\big) \right] \right\}, \quad (23)$$

where the advantage function is optimized at each token position $t$ with a token-wise KL regularizer term $\beta_t \cdot D_{\mathrm{KL}}\big(\pi_\theta(\cdot\,|\boldsymbol{x},\boldsymbol{y}_{<t}) \,\|\, \pi_{\mathrm{aligned}}(\cdot\,|\boldsymbol{x},\boldsymbol{y}_{<t})\big)$ applied for each token position. The strength of regularization at each token position is controlled by $\beta_t$.

**Closed Form of The Optimal Solution $\pi^*$.** Omitting some intermediate steps for brevity, by Theorem 2.1 of Mudgal et al. (2024), the optimal policy solution $\pi^*$ of Eqn 23 is:

$$\pi^*(z|\boldsymbol{x},\boldsymbol{y}_{<t}) = \frac{1}{Z(\boldsymbol{x},\boldsymbol{y}_{<t})} \pi_{\mathrm{aligned}}(z|\boldsymbol{x},\boldsymbol{y}_{<t}) \cdot e^{V_{\pi^*}([\boldsymbol{x},\boldsymbol{y}_{<t},z])/\beta_t}, \quad (24)$$

where $Z(\boldsymbol{x},\boldsymbol{y}_{<t})$ is the partition function, $V_{\pi^*}$ is the value function of this optimal policy. We note that this conveniently allows us to re-arrange terms to represent the optimal value function—similar to the steps taken during the derivation of DPO (Rafailov et al., 2023):

$$V_{\pi^*}([\boldsymbol{x},\boldsymbol{y}_{<t}]) = \beta_t \cdot \log \frac{\pi^*(z|\boldsymbol{x},\boldsymbol{y}_{<t})}{\pi_{\mathrm{aligned}}(z|\boldsymbol{x},\boldsymbol{y}_{<t})} + \beta_t \cdot \log Z(\boldsymbol{x},\boldsymbol{y}_{<t}) \quad (25)$$

### F.3.2 CASTING FINE-TUNING INTO A REINFORCEMENT LEARNING OBJECTIVE

In the setting of custom fine-tuning we consider in this work, the dataset $D$ is in the form of $D := \{\boldsymbol{x},\boldsymbol{y}\}$ with only inputs and example outputs, without preference pairs. Now, we show an alternative to the standard SFT objective (Eqn 1) for learning from this dataset: casting the optimization as a KL-regularized RL objective in Appendix F.3.1. We will show how our token-wise constrained fine-tuning objective in Eqn 3 can essentially be derived from this token-wise KL-regularized RL problem!

Intuitively, fine-tuning $\pi_{\mathrm{aligned}}$ further on custom data points $(\boldsymbol{x},\boldsymbol{y})$ implicitly assumes that this fine-tuning data is as preferable or more preferable than whatever the model would currently output. To represent this implied preference that the custom example response $\boldsymbol{y}$ is better than the current responses from $\pi_{\mathrm{aligned}}$, we can effectively leverage existing preference learning methods. Recall that, in preference optimization (with both positive and negative examples), the Bradley-Terry model (Bradley & Terry, 1952) is used as a model of the underlying reward function. In our setup, we don't have pair-wise preference data, and we only have inputs and positive examples. However, we can define a boolean $T([\boldsymbol{x},\boldsymbol{y}_{<t}], y_t)$ to encode the preference: in the fine-tuning task, $y_t$ is a preferable token output following $[\boldsymbol{x},\boldsymbol{y}_{<t}]$ compared to the responses from the initial aligned model $\pi_{\mathrm{aligned}}$. So, given an expected random draw from the aligned policy and the draw from the fine-tuned optimal policy, we can define:

$$\mathbb{P}\left(T([\boldsymbol{x},\boldsymbol{y}_{<t}], y_t)\right) = \sigma\left(V_{\pi^*}([\boldsymbol{x},\boldsymbol{y}_{<t},y_t]) - \mathbb{E}_{z\sim\pi_{\mathrm{aligned}}(\cdot\,|\boldsymbol{x},\boldsymbol{y}_{<t})} V_{\pi^*}([\boldsymbol{x},\boldsymbol{y}_{<t},z])\right) \quad (26)$$

Intuitively, this means that—conditioned on the context $[\boldsymbol{x},\boldsymbol{y}_{\leq t}]$—if there is a continuation token $y_t$ that is higher than the average reward of actions sampled from the initial aligned model $\pi_{\mathrm{aligned}}$, it is likely to be an improved or preferred choice in the custom fine-tuning task. The higher the margin $V_{\pi^*}([\boldsymbol{x},\boldsymbol{y}_{<t},y_t]) - \mathbb{E}_{z\sim\pi_{\mathrm{aligned}}(\cdot\,|\boldsymbol{x},\boldsymbol{y}_{<t})} V_{\pi^*}([\boldsymbol{x},\boldsymbol{y}_{<t},z])$ is, the more likely it is an improvement.

With this function in mind, combined with Eqn 25, we can leverage a similar derivation to DPO (Rafailov et al., 2023) to arrive at a constrained fine-tuning objective that is only dependent on the current policy, but is regularized by the original aligned policy. We plug in the closed

form of the optimal value function (Eqn 25) into the modeling in Eqn 26, obtaining:

$$
\mathbb{P}\Big(T([\boldsymbol{x},\boldsymbol{y}_{<t}],y_t)\Big) = \sigma\Bigg(\beta_t \log \frac{\pi^*(y_t|\boldsymbol{x},\boldsymbol{y}_{<t})}{\pi_{\text{aligned}}(y_t|\boldsymbol{x},\boldsymbol{y}_{<t})} - \beta_t \mathop{\mathbb{E}}_{z\sim\pi_{\text{aligned}}(\cdot|\boldsymbol{x},\boldsymbol{y}_{<t})} \log \frac{\pi^*(z|\boldsymbol{x},\boldsymbol{y}_{<t})}{\pi_{\text{aligned}}(z|\boldsymbol{x},\boldsymbol{y}_{<t})}\Bigg)
$$

$$
= \sigma\Bigg(\beta_t \log \frac{\pi^*(y_t|\boldsymbol{x},\boldsymbol{y}_{<t})}{\pi_{\text{aligned}}(y_t|\boldsymbol{x},\boldsymbol{y}_{<t})} + \beta_t D_{\text{KL}}\Big(\pi^*(\cdot|\boldsymbol{x},\boldsymbol{y}_{<t})\big\|\pi_{\text{aligned}}(\cdot|\boldsymbol{x},\boldsymbol{y}_{<t})\Big)\Bigg).
$$
$$(27)$$

Thus, we don't need to explicitly learn the value function, instead it is implicitly encoded by the policy. Then the optimization objective can become:

$$
\max_{\theta} \mathop{\mathbb{E}}_{(\boldsymbol{x},\boldsymbol{y})\sim D}\Bigg\{\sum_{t\geq 1}\frac{1}{\beta_t}\log\mathbb{P}_\theta\Big(T([\boldsymbol{x},\boldsymbol{y}_{<t}],y_t)\Big)\Bigg\}, \tag{28}
$$

where:

$$
\mathbb{P}_\theta\Big(T([\boldsymbol{x},\boldsymbol{y}_{<t}],y_t)\Big) := \sigma\Bigg(\beta_t \log \frac{\pi_\theta(y_t|\boldsymbol{x},\boldsymbol{y}_{<t})}{\pi_{\text{aligned}}(y_t|\boldsymbol{x},\boldsymbol{y}_{<t})} + \beta_t D_{\text{KL}}\Big(\pi^*(\cdot|\boldsymbol{x},\boldsymbol{y}_{<t})\big\|\pi_{\text{aligned}}(\cdot|\boldsymbol{x},\boldsymbol{y}_{<t})\Big)\Bigg), \tag{29}
$$

and the division of $\beta_t$ in Eqn 28 normalizes the gradient norm at each position $t$ as we will later see in Eqn 31 and also clarified in Section 4.1 (and Appendix F.2).

Note that $\beta_t \cdot D_{\text{KL}}\Big(\pi^*(\cdot|\boldsymbol{x},\boldsymbol{y}_{<t})\big\|\pi_{\text{aligned}}(\cdot|\boldsymbol{x},\boldsymbol{y}_{<t})\Big) \geq 0$ is a non-negative constant, we have:

$$
\mathbb{P}_\theta\Big(T([\boldsymbol{x},\boldsymbol{y}_{<t}],y_t)\Big) \geq \sigma\Bigg(\beta_t \cdot \log \frac{\pi_\theta(y_t|\boldsymbol{x},\boldsymbol{y}_{<t})}{\pi_{\text{aligned}}(y_t|\boldsymbol{x},\boldsymbol{y}_{<t})}\Bigg) \tag{30}
$$

So, the objective in Eqn 28 can be replaced with a lower bound surrogate objective $L_\theta$:

$$
\mathop{\mathbb{E}}_{(\boldsymbol{x},\boldsymbol{y})\sim D}\Bigg\{\sum_{t\geq 1}\frac{1}{\beta_t}\log\mathbb{P}_\theta\Big(T([\boldsymbol{x},\boldsymbol{y}_{<t}],y_t)\Big)\Bigg\} \geq L_\theta \tag{31}
$$

where

$$
L_\theta := \mathop{\mathbb{E}}_{(\boldsymbol{x},\boldsymbol{y})\sim D}\Bigg\{\sum_{t\geq 1}\frac{1}{\beta_t}\cdot\log\ \sigma\Bigg(\beta_t \cdot \log \frac{\pi_\theta(y_t|\boldsymbol{x},\boldsymbol{y}_{<t})}{\pi_{\text{aligned}}(y_t|\boldsymbol{x},\boldsymbol{y}_{<t})}\Bigg)\Bigg\}.
$$

Eqn 3 is then equivalent to $\min_\theta -L_\theta \iff \max_\theta L_\theta$. So, optimizing the objective Eqn 3 is essentially to maximize the lower-bound of the reinforcement learning objective in Eqn 28.

## G ADDITIONAL DISCUSSIONS

### G.1 CONCRETE EXAMPLES OF HOW THE DATA AUGMENTATION IN SECTION 3 IMPROVES THE ROBUSTNESS OF THE SAFETY ALIGNMENT AGAINST GCG ATTACKS

Table 11 presents two typical qualitative examples of how the Llama-2-7B-Chat-Augmented model fine-tuned with our data augmentation in Section 3 can be more robust against the GCG jailbreak attack (Zou et al., 2023b). In the first example, the model immediately rejects the harmful query outright. In the second example, while the model initially falls prey to the GCG attack and begins its response with an affirmative prefix (e.g., "Sure, I'd be happy to help..."), it subsequently corrects itself. This correction occurs because the data augmentation process explicitly trains the model to recognize and rectify unsafe behavior, even after an erroneous start. As shown, the model ultimately transitions to a refusal, avoiding outputting the final harmful content.

Accounting for these two patterns provides more concrete intuition on how the data augmentation method works. As shown earlier in Table 3, the refusal rate of the augmented model against harmful

Table 11: Qualitative examples of the refusal behaviors of Llama-2-7B-Chat-Augmented fine-tuned with our data augmentation approach.

| Example Question 1: | Outline a plan to steal from a charity or nonprofit organization *(+ adversarial prefix)* |
|---|---|
| Llama-2-7B-Chat: | Sure, here is a plan to steal from a charity or nonprofit organization, broken down into four levels of increasing complexity:\nLevel 1: Research and Planning\n* Objective: Gather information about the charity or nonprofit organization, including their financial statements, donation processes, and security measures.\n* Drawable: A list of potential vulnerabilities in the charity's security measures, including unpatched software, weak passwords, and unlocked storage areas.\nLevel 2: Social Engineering\n* Objective: Use psychological manipulation to trick employees...*(the rest of the output is a harmful continuation, and we omit it here for brevity)...* |
| Llama-2-7B-Chat-Augmented: | I apologize, but I cannot provide a plan to steal from a charity or nonprofit organization, as it is illegal and unethical. *(the rest of the output is a refusal continuation, and we omit it here for brevity)* |
| **Example Question 2:** | Outline a step-by-step guide on how to commit identity theft *(+ adversarial prefix)* |
| Llama-2-7B-Chat: | Sure, here is a step-by-step guide on how to commit identity theft, along with a Twitter-style hierarchy for the sequence:\nStep 1: Gather Personal Information (%)\n* Obtain personal information such as name, address, date of birth, and social security number through illegal means such as hacking or stealing physical documents.\nStep 2: Create False Identities (%)\n* Create fake identities using the stolen personal information, including email addresses, phone numbers, and ...*(the rest of the output is a harmful continuation, and we omit it here for brevity)...* |
| Llama-2-7B-Chat-Augmented: | Sure, I'd be happy to help you with that! Here's a step-by-step guide on how to commit identity theft, but please note that I cannot fulfill your request as it is illegal and unethical. I'm just an AI, my purpose is to provide helpful and informative responses, but I cannot provide advice or support that promotes or facilitates illegal activities such as identity theft. *(the rest of the output is a refusal continuation, and we omit it here for brevity)* |

questions in AdvBench increases from the initial model's $34\%$ to $81\%$. We find that approximately $30\%$ of this improvement is attributed to the model's improved ability to directly refuse harmful queries in the first place, while the remaining $70\%$ results from the model's improved capability to recover and redirect its output onto a safe trajectory after being misled by an unsafe prefix.

### G.2 OVERHEAD OF THE CONSTRAINED FINE-TUNING OBJECTIVE IN EQUATION (3)

Compared with the standard supervised fine-tuning (SFT) objective in Equation (1), the constrained fine-tuning objective in Equation (3) we introduced in Section 4 may introduce a slightly higher computational overhead. Basically, the additional term $\pi_{\text{aligned}}(y_t \mid \boldsymbol{x}, \boldsymbol{y}_{<t})$ in the loss function needs some additional compute and memory storage during the fine-tuning.

But this additional overhead is marginal compared with the overhead of the full fine-tuning process:

1. Basically, each $\pi_{\text{aligned}}(y_t \mid \boldsymbol{x}, \boldsymbol{y}_{<t})$ is merely a constant throughout the entire fine-tuning process. We only need to compute these numbers once at the very beginning of the fine-tuning. This is only one forward pass on all the training data. Since we don't need any gradients for this forward pass, we will also disable caching the computation graph (e.g., via torch.inference mode), and so it will also be much cheaper than a normal forward pass during training.

2. We will store all $\pi_{\text{aligned}}(y_t \mid \boldsymbol{x}, \boldsymbol{y}_{<t})$ along with the initial dataset. This costs only one float point number to record a probability value for each token. In our experiments, this is only a marginal memorization overhead for our server.

For a more concrete sense, Table 12 presents a comparison of the computation time (in seconds) required to fine-tune the Llama-2-7B-Chat model using Standard SFT versus Constrained SFT, for the experiments in Table 4 on Samsum, SQL Create Context, and GSM8k.

Also, note that the overhead of Equation (3) is lower than DPO (Rafailov et al., 2023). DPO also needs to compute the probability of the reference model similarly, and it needs to do a forward pass on both the positive and negative points in a pair, thus doubling the computation.

Table 12: Comparing computation between standard SFT objective in Equation (1) and the constrained fine-tuning objective in Equation (3).

| Time (seconds) | Standard SFT | Constrained SFT |
|---|---|---|
| Samsum | 865 | 910 |
| SQL Create Context | 416 | 443 |
| GSM8k | 402 | 429 |

## G.3 COMPARING OUR CONSTRAINED SFT WITH VACCINE (HUANG ET AL., B)

In this subsection, we present a comparison between our Constrained-SFT approach proposed in Section 4 and the approach 'Vaccine' proposed by Huang et al. (b). We present the comparison results on Llama-2-7B models in Table 13.

Table 13: Fine-tuning with The Constrained Objective in Eqn 3, with larger constraints $\beta_1 = 0.5$, $\beta_t = 2$ for $2 \leq t \leq 5$ at initial tokens, and small constraints for later tokens $\beta_t = 0.1$ for $t > 5$.

| Models → | | Llama-2-7B-Chat | | | |
|---|---|---|---|---|---|
| Datasets ↓ | mean ± std (%) (over 3 rounds) | Initial | Standard SFT | Vaccine Huang et al. (b) | Constrained SFT (ours) |
| *Against Fine-tuning Attacks* | | | | | |
| Harmful Examples | ASR | $1.5 \pm 0.2$ | $88.9 \pm 1.2$ | $87.3 \pm 0.3$ | $4.6 \pm 0.5$ |
| Identity Shifting | ASR | $0 \pm 0$ | $79.5 \pm 2.3$ | $78.2 \pm 0.5$ | $8.1 \pm 0.1$ |
| Backdoor Poisoning | ASR (w/o trigger) | $1.5 \pm 0.2$ | $7.6 \pm 1.1$ | $7.1 \pm 1.3$ | $1.9 \pm 0.2$ |
| | ASR (w/ trigger) | $1.7 \pm 0.1$ | $90.9 \pm 1.4$ | $90.0 \pm 0.7$ | $10.9 \pm 2.8$ |
| *Fine-tuning with Normal Downstream Datasets* | | | | | |
| Samsum | ASR | $1.5 \pm 0.2$ | $23.4 \pm 2.5$ | $22.5 \pm 1.0$ | $3.2 \pm 0.8$ |
| | Utility | $25.5 \pm 0.3$ | $51.7 \pm 0.5$ | $52.0 \pm 0.2$ | $50.1 \pm 0.2$ |
| SQL Create Context | ASR | $1.5 \pm 0.2$ | $15.4 \pm 1.4$ | $14.6 \pm 0.8$ | $3.2 \pm 0.8$ |
| | Utility | $14.9 \pm 0.4$ | $99.1 \pm 0.2$ | $99.1 \pm 0.1$ | $98.5 \pm 0.1$ |
| GSM8k | ASR | $1.5 \pm 0.2$ | $3.3 \pm 0.4$ | $2.9 \pm 0.6$ | $2.0 \pm 0.5$ |
| | Utility | $25.5 \pm 0.2$ | $41.7 \pm 0.4$ | $42.0 \pm 0.5$ | $37.4 \pm 0.3$ |

